# GULF18, a high-resolution NEMO-based tidal ocean model of the Arabian/Persian Gulf

Diego Bruciaferri[1], Marina Tonani[1,*], Isabella Ascione[1], Fahad Al Senafi[2], Enda O'Dea[1], Helene T. Hewitt[1], and Andrew Saulter[1]

[1]Met Office, Exeter, EX1 3PB, UK
[2]Department of Marine Science, Kuwait University, Safat, Kuwait
[*]*now at Mercator Ocean International, Toulouse, France*

**Correspondence:** Diego Bruciaferri (diego.bruciaferri@metoffice.gov.uk)

**Abstract.** The sensitivity of a shelf sea model of the Gulf area to changes in the bathymetry, lateral and vertical resolution, vertical coordinates and external forcing is explored. Two new Gulf models with a resolution of 1.8 km named GULF18-3.6 and GULF18-4.0 differing only in the vertical coordinate system and the NEMO codebase employed (NEMO-3.6 and NEMO-4.0.4, respectively) are introduced. We compare them against the existing 4 km PGM4 model, which is based on NEMO-3.4 and is developed and used by the Met Office. PGM4 and GULF18-3.6 use a similar type of quasi-terrain-following vertical levels while GULF18-4.0 employs the multi-envelope method to discretise the model domain in the vertical direction. Our assessment compares non-assimilative hindcast simulations of the three Gulf models for the period 2014-2017 against available observations of tides, hydrography and surface currents. Numerical results indicate that both high resolution models have higher skill than PGM4 in representing the sea surface temperature and the water column stratification on the shelf. In addition, in the proximity of the shelf-break and the deep part of the domain GULF18-4.0 generally presents the highest accuracy, demonstrating the benefit of optimising the vertical grid for the leading physical processes. For the surface currents, the three models give comparable results on the shelf, while the higher resolution models might be prone to the double penalty effect in deeper areas. For the tides, PGM4 has a better skill than GULF18 models and our tidal harmonic analysis suggests that future work may be needed in order to get real benefit from using a more realistic bottom topography as in the case of the GULF18 models.

## 1 Introduction

The Arabian/Persian Gulf (hereafter, "Gulf") is a shallow, semi-enclosed sea located between the Arabian Peninsula and the south west of Iran and connected to the open Indian ocean via the Strait of Hormuz, the Gulf of Oman and the Arabian Sea. It is an elongated shelf sea representing the main supply of water for industrial and domestic usage for all its surrounding countries. The Gulf region can be impacted by various natural and anthropogenic factors that can importantly affect the quality of its waters and influence the equilibrium of its marine ecosystem (Richlen et al., 2010; Al Shehhi et al., 2014; Zhao and Ghedira, 2014; Gherboudj and Ghedira, 2014). For example, the Gulf area represents one of the major oil-rich regions of the World, where the risk for oil spills and illicit discharges with potential adverse ecological impacts is extremely high (Essa et al.,

2005; Zhao et al., 2014, 2015). Therefore, it is of fundamental importance to understand and accurately predict the short-term dynamics and state of the waters of this basin as well as its climatic and anthropogenic-induced variability.

The Gulf dynamics arise from complex interactions between topography, atmospheric fluxes of heat (Al Senafi et al., 2019), freshwater and momentum, river discharges and tides (see for example the exhaustive review given by Hyder et al. (2013)). A general wind- and buoyancy-driven cyclonic inverse estuarine circulation transports low salinity water originating from the Arabian sea and primarily entering the basin from the northern part of the Strait of Hormuz towards the northwestern and southeastern areas of the Gulf (Reynolds, 1993; Johns et al., 1999; Al Senafi and Anis, 2020b). Here, the combination of large evaporation and shallow depths leads to the formation of highly saline waters which leave the Gulf through the deep part of the Strait of Hormuz forming dense bottom waters cascading at the shelf-break (e.g., Shapiro et al. (2017)). Tidal currents are strong and important in controlling the stratification and fronts formation of the basin (particularly close to the Strait of Hormuz, e.g. Matsuyama et al. (1998); Pous et al. (2013); Li et al. (2020)). Whilst early modelling studies found the tidal residual flow to be weak and to not contribute significantly to the main circulation of the Gulf (e.g. Pous et al. (2013)), more recent numerical efforts (e.g. Mashayekh Poul et al. (2016)) showed strong tide-induced residual currents of the order $\approx 15$ $cm\ s^{-1}$ in the Strait of Hormuz, more than 5 times greater than previous studies.

Few 3D numerical models of the Gulf hydrodynamics exist in literature. For example, Pous et al. (2015) used a 9 $km$ resolution regional implementation of the MARS3D model (Lazure and Dumas, 2008) with 30 terrain-following $\sigma$-levels to describe the intraseasonal to interannual variability in the Gulf circulation and exchange through the Strait of Hormuz. Similarly, Al Azhar et al. (2016) implemented the Regional Ocean Modeling System (ROMS, Shchepetkin and McWilliams (2005)) in the Gulf region with a resolution of 5 $km$ and 25 terrain-following $s$-levels to study the sensitivity of the model to different vertical turbulence mixing and light penetration schemes. Hyder et al. (2013) presented and evaluated the forecasting skills of PGM4, a regional tidal implementation of the Nucleus for European Models of the Ocean (NEMO) (Madec and NEMO-team, 2016) numerical code with a horizontal resolution of 4 km and 31 $s$-levels. Shapiro et al. (2017) used the NEMO ocean model with a resolution of $\approx 1.8\ km$ and 52 hybrid $s$-$z$ levels (Shapiro et al., 2013) to characterise the seasonal variability of the dense outflow from the Gulf into the Gulf of Oman. Likewise, Vasou et al. (2020) used NEMO with a resolution of $\approx 2.6\ km$ and 50 $z$-levels with partial steps to study the variability of the water mass exchange between the Gulf and the Indian Ocean. Recently, Lorenz et al. (2020) applied the General Estuarine Transport Model (GETM; Klingbeil and Burchard (2013)) with a resolution of $\approx 1.8\ km$ and 40 adaptive vertical layers (Hofmeister et al., 2010) to investigate the properties of the exchange flow of the Gulf.

In this paper we describe and assess GULF18, a new 1.8 km resolution tidal ocean model of the Gulf area, against observations and PGM4, the model developed and used by the Met Office (Hyder et al., 2013). The aim of this study is to explore the impact of using a more realistic bottom topography and coastline, increasing the lateral and vertical resolution, optimising the vertical discretisation scheme for the leading physical processes and updating the external forcing on the accuracy of a Gulf

model.

60

Some of these developments were motivated by the lessons learned from the operational ocean forecasting system that the Met Office runs for the north-west European shelf (NWS). For example, Graham et al. (2018a, b) and Tonani et al. (2019) showed that resolving the internal Rossby radius both on the shelf and in the deep ocean improves the accuracy of the simulated mesoscale dynamics, better resolving important circulation patterns of the NWS such as the European slope currents or the across shelf transport. Similarly, Siddorn and Furner (2013) and O'Dea et al. (2017) demonstrated the importance of increasing the vertical resolution, especially in the case of haline ocean fronts in shallow, tidally-mixed areas or for the fluxes at the sea surface. More recently, Bruciaferri et al. (2020) and Wise et al. (2021) proved that using a multi-envelope $s$-coordinate system (Bruciaferri et al., 2018) allows significant improvements in the accuracy of an ocean model including shelf and deep ocean areas when compared to traditional models employing geopotential or terrain-following levels.

70

At the time when the model development was carried out, the latest available stable version of the NEMO code was v4.0.4 (Madec and NEMO-team, 2019). This new version of the code differs significantly from v3.4 (Madec and NEMO-team, 2012), the release used by PGM4. For this reason, two GULF18 models mainly differing in the NEMO code version and the vertical discretisation scheme are developed and compared to PGM4 in this study, allowing a better understanding and assessment of the impact of each model development included in this new configuration.

Freely accessible observations of the water column physical properties for oceanographic research purposes are scarce in the Gulf (Hyder et al., 2013) and extensive efforts have been spent in this study to gather all the possible available observations to validate the skill of our models. To the best of our knowledge, freely accessible and reasonably recent (collected during the last decade) datasets cover the 2014-2017 span. While such a period is not long enough to evaluate the skill of our models on climatic time scales, it is suitable to assess their ability in predicting the short term variability of the Gulf dynamics.

The paper is organised as follows. Section 2 details the main features of the new GULF18 model, highlighting the key differences with the existing PGM4 model in the model domain geometry (Sec. 2.1), vertical discretization schemes (Sec. 2.2), model core and physics (Sec. 2.3) and external forcing and initialisation (Sec. 2.4). After, Sec. 3 describes the methodology to assess models' skill while Sec. 4 presents and discusses our main results for tides (Sec. 4.1), sea surface temperature (Sec. 4.2), water column stratification (Sec. 4.3) and sea surface currents (Sec. 4.4). Finally, Sec. 5 summarises our main conclusions and future development's plans.

## 2 GULF18 ocean model

90 In this study, two different GULF18 models are developed and compared to the existing PGM4. Both GULF18 configurations share the same bathymetry and horizontal grid but they differ in the vertical discretisation scheme and the version of the NEMO

| MODEL PARAMETERS | | PGM4 | GULF18-3.6 | GULF18-4.0 |
|---|---|---|---|---|
| **DOMAIN, HORIZONTAL GRID AND BATHYMETRY (SEC. 2.1)** | | | | |
| **Bathymetry** | | GEBCO 2008 + ad-hoc modific. | GEBCO 2014 | |
| **Horizontal grid** | *Number of points* | $244 \times 172$ | $584 \times 432$ | |
| | *Resolution [km]* | $\approx 4.3$ | $\approx 1.8$ | |
| **Land-Sea mask & coastline** | | from bathymetry + ad-hoc modific. | from bathymetry | |
| **VERTICAL DISCRETISATION (SEC. 2.2)** | | | | |
| **Discretisation scheme** | | vanishing quasi-sigma Song and Haidvogel (1994) | vanishing quasi-sigma Siddorn and Furner (2013) | multi-envelope Bruciaferri et al. (2018) |
| **Number of levels** | | 31 | 52 | |
| **Surface level thickness[m]** | | $\approx 0.3 - 6$ | 1 | |
| **DYNAMICAL CORE AND PHYSICS (SEC. 2.3)** | | | | |
| **NEMO code** | | v3.4-stable | v3.6-stable | v4.0.4 |
| **Reference density** | $[kg\ m^{-3}]$ | 1027 | 1020 | 1026 |
| **Lateral SGP** | *Harmonic diffusivity* $[m^2 s^{-1}]$ | 3D-constant 50 | Smagorinsky-like $1 - 30$ | 3D-constant 2 |
| | *Bi-harmonic viscosity* $[m^4 s^{-1}]$ | 3D-constant $-1.0 \times 10^{10}$ | 3D-constant $-4.5 \times 10^8$ | Mesh size and depth dependent $[-3.84 \times 10^8, -4.54 \times 10^8]$ |
| **Penetrative solar radiation** | | POLCOMS fixed length scale formulation | NEMO RGB formulation | |
| **EXTERNAL FORCING (SEC. 2.4)** | | | | |
| **Surface boundary conditions** | | Flux formulation | Large and Yeager (2009) BULK formulae | |
| **Tidal forcing** | | TPXOv7.2 | FES2014 | |
| **Rivers** | | 1 | 5 | |

**Table 1.** Summary of the differences between the currently operational PGM4 configuration and GULF18-3.6 and GULF18-4.0 ocean models.

code employed. The first model, named GULF18-3.6, uses NEMO v3.6 (Madec and NEMO-team, 2016) and vanishing quasi-sigma (VQS) vertical levels (Dukhovskoy et al., 2009) similarly to PGM4. The second configuration, named GULF18-4.0, is based on NEMO v4.0.4 (Madec and NEMO-team, 2019) and employs the multi-envelope (ME) method (Bruciaferri et al., 95  2018) to discretise the domain in the vertical direction. Table 1 summarises the main differences between both GULF18 configurations (hereafter GULF18-*) and PGM4. In the next sections, the key components and parametrizations of GULF18-* along with their main differences with PGM4 are outlined and discussed.

## 2.1 Domain, horizontal grid and bathymetry

GULF18-* and PGM4 configurations cover the same area extending from $47° \, 36'$ E to $57° \, 38'$ E and from $23° \, 03'$ N to $30° \, 30'$ N in the zonal and meridional directions, respectively. In addition, they also share the same single open boundary with the adjacent Indian Ocean located in the Gulf of Oman (see Fig. 2a and b).

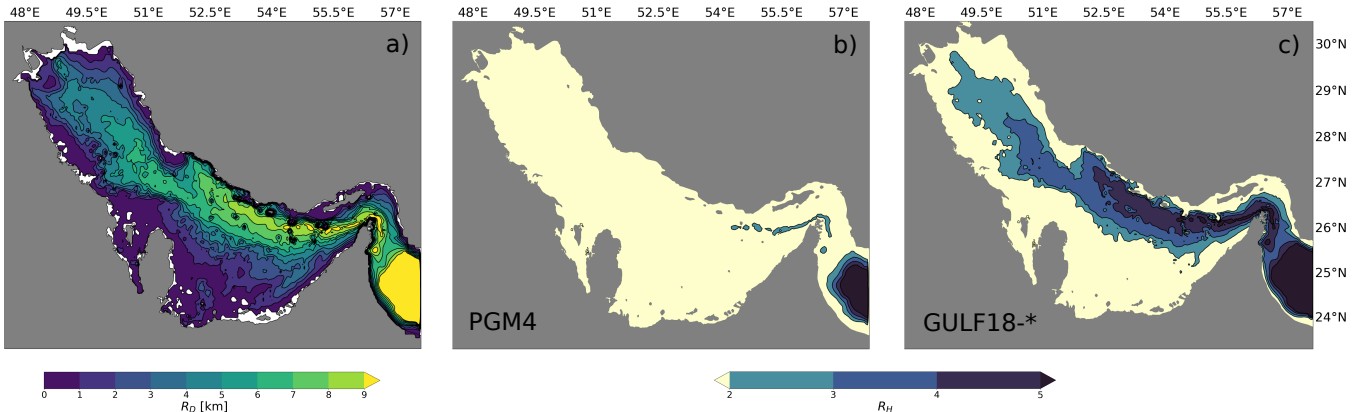

**Figure 1.** First baroclinic Rossby radius of deformation $R_D$ in the Gulf (a) and maps of the distribution of the Hallberg (2013) metric $R_H$ in the case of PGM4 (b) and GULF18-* (c) configurations.

Both GULF18 configurations and PGM4 implement a regular geographical horizontal grid with grid lines aligned with parallels and meridians. However, PGM4 uses $244 \times 172$ grid points in the zonal and meridional directions, respectively, corresponding to a nominal lateral resolution of $\approx 4 \, km$, while GULF18-3.6 and GULF18-4.0 discretise the horizontal domain with $584 \times 432$ grid points, achieving a nominal resolution of $\approx 1.8 \, km$. Figure 1a presents a map of the first baroclinic Rossby radius of deformation $R_D = c_1/|f|$ in the Gulf, with $f$ the Coriolis parameter, $c_1$ the first eigenvalue satisfying the boundary value problem for the vertical velocity (Chelton et al., 1998) and where 2013-2018 PGM4 averaged temperature and salinity fields have been used in the computation of $c_1$. Hallberg (2013) proposed the metric $R_H = R_D/\sqrt{(\Delta x^2 + \Delta y^2)/2}$ as an appropriate measure of whether the baroclinic eddy dynamics are likely to be well resolved by a model with horizontal grid spacing $\Delta x$ and $\Delta y$. Typically, a model is defined 'eddy-permitting' when $R_H < 2$ while it is considered 'eddy-resolving' when $R_H > 2$ (e.g., Hallberg (2013); Sein et al. (2018); Yankovsky et al. (2022)). Models which are not fully eddy-resolving but where areas with $R_H > 2$ represent an important part of the domain are often classified as 'eddy-rich' (e.g., Fox-Kemper and Bachman (2014); Moreton et al. (2020)). Figures 1b and c present the distribution of $R_H$ in the case of PGM4 and GULF18-* models, respectively, showing that PGM4 can be classified as an eddy-permitting model while GULF18-* can be considered eddy-rich configurations.

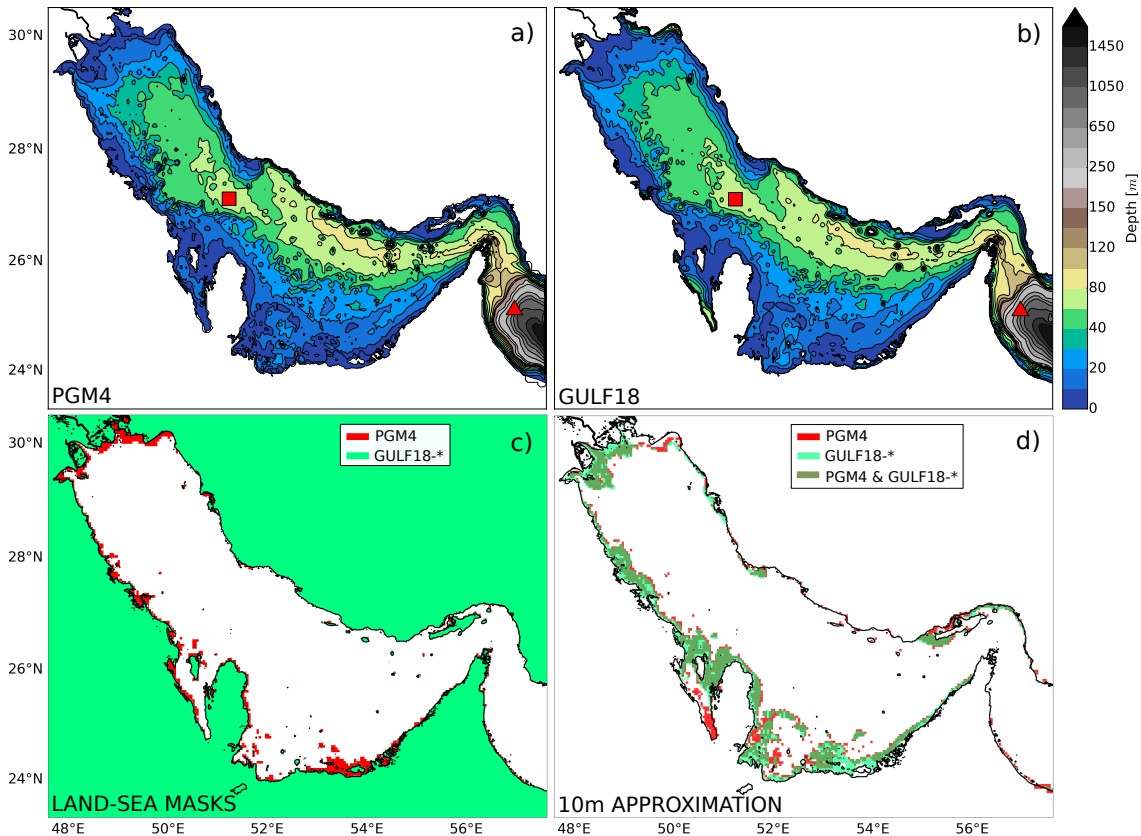

**Figure 2.** Upper row: bottom topography of PGM4 (a) and GULF18-* (b) models; Bottom row: Land-sea mask (c) and grid cells where the 10 m approximation has been applied (d) only in PGM4 (red), only in GULF18-* (light green) or in both configurations (dark green). The red square and triangle in panel a and b represent the location where the profiles of models' levels vertical distribution shown in Fig. 5 are extracted.

In both GULF18 configurations the bottom topography $H(x,y)$ (with $x$ and $y$ representing the zonal and meridional directions, respectively) is computed from the 30 arc-second resolution General Bathymetric Chart of the Oceans (GEBCO) 2014 dataset (see Fig. 2b). In the deep part of the domain (depth $> 300\ m$) GULF18-* bathymetry is merged with the bottom topography of the Met Office GO6 global ocean configuration at $1/12°$ resolution ((Storkey et al., 2018)). Conversely, the PGM4 model bathymetry is based on the GEBCO 2008 dataset with some additional smoothing to alleviate horizontal pressure gradient errors with terrain-following vertical coordinates (especially in the proximity of the shelf-break, see Fig. 2a) and ad-hoc modifications to widen the channels around Bahrain and the Gulf of Salwa in order to minimise salinity drift due to evaporation (Hyder et al., 2013).

In order to deal with the large tidal excursion characterising the Gulf area, GULF18-* and PGM4 models apply the same strategy of setting the minimum depth of their bottom topography to $10\ m$ - i.e. the model bathymetry is modified deepening

to $10\ m$ at every grid point where the original depth is shallower than this threshold. Such a crude modelling choice represents the only available solution when tidal ranges are large but the numerical model employed has no wetting-and-drying capability. Figure 2c presents the land-sea mask of GULF18-* (in green) and PGM4 (in red) ocean models while Fig. 2d illustrates the areas where the minimum depth parameterization is applied only in PGM4 (in red), only in GULF18-* (in light green) or in both ocean configurations (in dark green). Both figures clearly show that in the case of PGM4 the land-sea mask and coastline significantly diverge from the original model bathymetry, including important ad-hoc modifications, especially in the northern and southern regions of the domain and in the proximity of Bahrain. In these areas, PGM4 model sets to land all those ocean grid points where the depth is $< 3\ m$. On the other hand, GULF18-* land-sea mask and model coastline perfectly agree with the original model bathymetry: this modelling choice was preferred since starting from v4.0 the NEMO code is equipped with a wetting-and-drying algorithm (O'Dea et al., 2020) which could be employed in the future to have a more realistic representation of the water level evolution.

## 2.2 Vertical discretisation

In the vertical direction, GULF18-3.6 and PGM4 models implement a vanishing quasi-sigma (VQS) vertical discretization scheme where computational surfaces follow an envelope bathymetry surface rather than the actual model bottom topography (Dukhovskoy et al., 2009; O'Dea et al., 2012). Such an envelope is computed smoothing the model bathymetry with the Martinho and Batteen (2006) algorithm to ensure that the maximum value of the slope parameter $r = |H_a - H_b|/(H_a + H_b)$, with $H_a$ and $H_b$ the depths of adjacent grid points (Mellor et al., 1998), is less than a given threshold $r_{max}$. This solution allows one to have computational surfaces that are less tilted than in pure terrain-following models, hence reducing the errors in computing horizontal pressure gradients (e.g. Shapiro et al. (2013); Bruciaferri et al. (2018)). However, since computational surfaces are no longer strictly terrain-following, model cells are masked out in those grid points where the envelope is deeper than the model bathymetry. As a result, when a too severe $r_{max}$ threshold is used, the model bathymetry can include 'saw-tooth' structures similar to z-level steps that can potentially affect the accuracy of the bottom boundary layer dynamics represented by the model, including cross-shelf cascading and tides (Bruciaferri et al., 2020; Wise et al., 2021).

Both GULF18-3.6 and PGM4 models use a gentle maximum slope parameter threshold $r_{max} = 0.3$ to generate their envelope bathymetry (see panels a and b of Fig. 3). In the case of GULF18-3.6, this $r_{max}$ value was chosen after sensitivity tests for horizontal pressure gradient errors (HPGE) and tidal dynamics accuracy. The HPGE test is a classical (e.g., Haidvogel and Beckmann (1999)) idealised numerical experiment where the model is initialised with no horizontal density gradients and neither external forcing nor explicit diffusion is applied. In this type of problem, the analytical solution for ocean currents is $0\ m\ s^{-1}$. However, when model levels are not aligned with geopotential surfaces finite difference mathematics may introduce errors in the computation of the pressure gradient force generating undesired numerical spurious currents (e.g., Mellor et al. (1998); Berntsen (2002)). GULF18-3.6 sensitivity tests showed that whilst decreasing the $r_{max}$ did not significantly reduce HPGE (after 30 days models using $r_{max}$ equal to 0.3 or 0.1 developed similar basin averaged spurious currents of $\approx 4\ cm\ s^{-1}$),

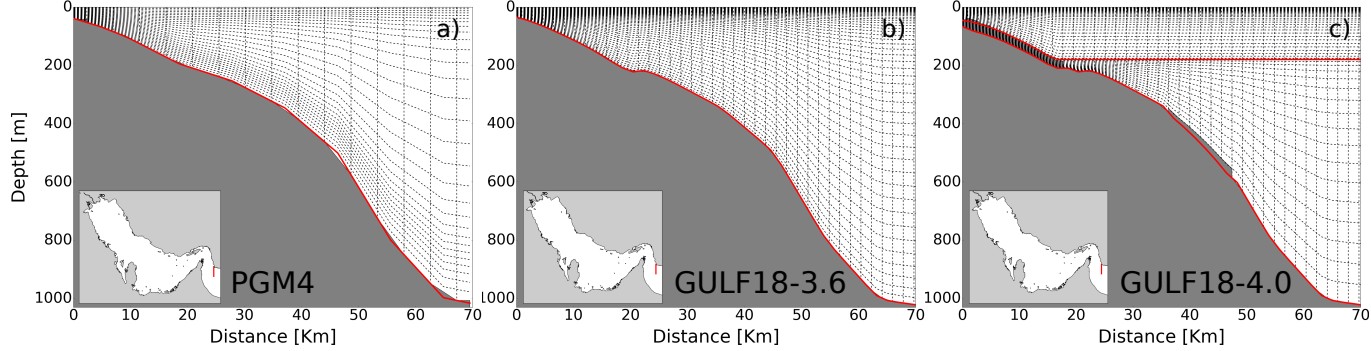

**Figure 3.** Example of cross-sections showing the model bathymetry and the numerical mesh of PGM4 (a), GULF18-3.6 (b) and GULF18-4.0 (c) configurations. In the case of PGM4 (a) and GULF18-3.6 (b), the red lines represent the envelope bathymetry; in the case of GULF18-4.0 (c), they identify the generalised upper and deeper envelopes $H_e^1(x,y)$ and $H_e^2(x,y)$.

using a more severe $r_{max}$ had a negative impact on the accuracy of the tidal dynamics represented by the model (e.g., the Mean Absolute Error of the simulated M2 tidal component increased by $\approx 1\ cm$ when the $r_{max}$ was reduced from 0.3 to 0.1).

165

GULF18-4.0 model discretises the vertical domain via a multi-envelope (ME) s-coordinate system (Bruciaferri et al., 2018). This is a generalised vertical coordinate system where model levels are curved and adjusted to arbitrarily defined surfaces (aka envelopes) rather than following geopotential levels, the actual bottom topography or a single envelope bathymetry, as is the case for the GULF18-3.6 and PGM4 models. In such a way, computational levels can be optimised for the leading dynamics
170 in different sub-domains of the model (see Bruciaferri et al. (2018, 2020) for the details).

In the case of a shelf sea model such as GULF18, the physical processes that a vertical grid should be able to accurately represent and prioritise are the strong tides and vertical mixing on the shelf, the cross-shelf transport and dense water cascading at the shelf-break and the turbulent exchanges with the atmosphere at the surface (Simpson and Sharples, 2012). Keeping this
175 in mind, the ME vertical grid of GULF18-4.0 is configured using 2 envelopes (see Fig. 3c):

- the upper envelope $H_e^1(x,y)$ follows the actual topography $H(x,y)$ from a minimum depth of $10\ m$ to a maximum depth of $180\ m$ and is smoothed via the Martinho and Batteen (2006) algorithm to have $r_{max} = 0.3$. With such an envelope, almost fully terrain-following computational surfaces are used where the bathymetry is shallower than 180 m while elsewhere the upper water column is discretised with geopotential model levels, allowing to minimise HPGE while
180 accurately representing mixed layer processes.

- the deeper envelope is computed as $H_e^2(x,y) = max\{H_e^1(x,y) + h, H(x,y)\}$, where $h = 30\ m$ represents a user-defined offset parameter, and the Martinho and Batteen (2006) smoothing algorithm is applied to ensure that $r_{max} = 0.1$. In this way, in areas where the bathymetry is deeper than $180\ m$ the model uses nearly terrain-following levels just in the proximity of the bottom topography while in the open ocean model levels relax toward geopotential surfaces.

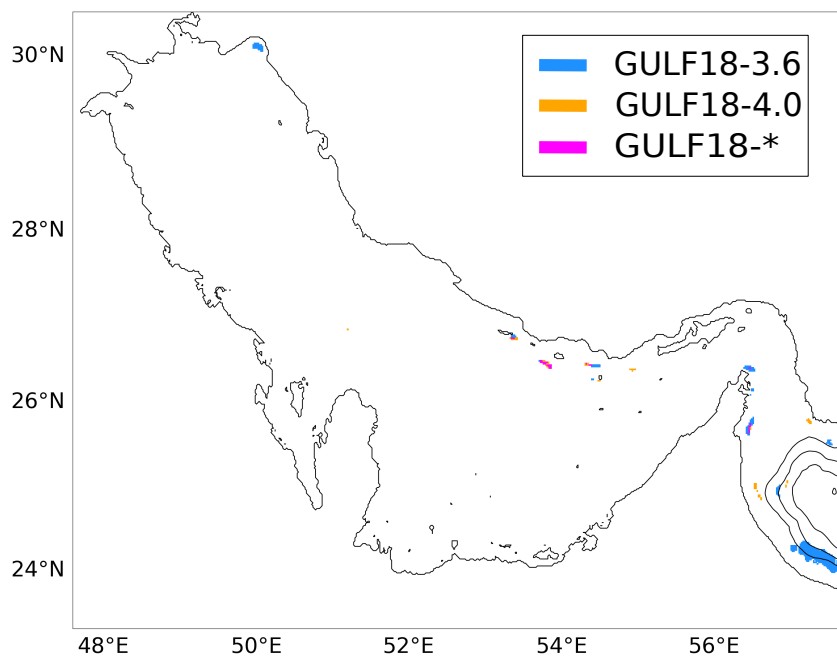

**Figure 4.** Grid points where spurious currents diagnosed via a HPGE test were $> 5\ cm\ s^{-1}$ only in GULF18-3.6 (blue), only in GULF18-4.0 (orange) or in both high resolution configurations (magenta).

Wise et al. (2021) showed that in a NWS model using a ME system with envelopes optimised to have HPGE $< 5\ cm\ s^{-1}$ gives significantly increased accuracy compared to VQS levels. Learning from this experience, both GULF18-4.0 envelopes were additionally smoothed at grid points where HPGE (assessed with a HPGE test) were larger than $6\ cm\ s^{-1}$ in the case of $H_e^1(x,y)$ and $3\ cm\ s^{-1}$ for $H_e^2(x,y)$, with target $r_{max}$ parameters equal to $0.09$ and $0.04$, respectively. In the case of the upper envelope, a less restrictive threshold is applied than Wise et al. (2021) since GULF18-3.6 sensitivity tests showed that the accuracy of the simulated tidal dynamics is highly sensitive to how the model represents the bottom topography.

Figure 4 shows the model cells where the maximum (in the vertical and time) spurious velocities were $> 5\ cm\ s^{-1}$ only in GULF18-3.6 (in blue), only in GULF18-4.0 (in orange) and in both GULF18-* configurations (in magenta) after a 30 days-long HPGE test. Numerical results show that the multi-envelope configuration chosen for GULF18-4.0 allows the use of a 3D varying $r_{max}$ parameter which reduces the large HPGE affecting GULF18-3.6 in the proximity of the continental slope while minimising the number of undesired artificial saw-tooth structures on the shelf and shelf-break.

Both GULF18-* configurations use 52 computational surfaces to discretise the vertical domain while PGM4 employs 31 model levels. Figure 5 presents the vertical resolution of PGM4 (in red), GULF18-3.6 (in green) and GULF18-4.0 (in blue) models at two representative locations of the shelf (a) and the deep basin (b), respectively. The vertical distribution of PGM4 computational surfaces is stretched according to the Song and Haidvogel (1994) function while GULF18-3.6 uses the Siddorn

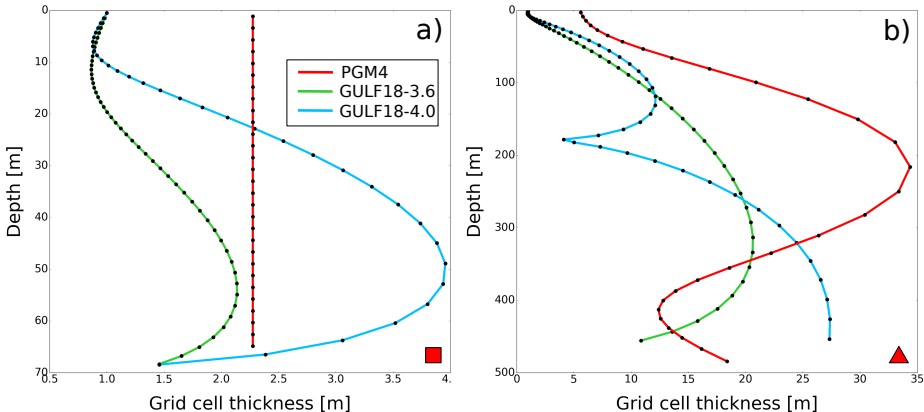

**Figure 5.** Model cell thickness as a function of depth for PGM4 (in red), GULF18-3.6 (in green) and GULF18-4.0 (in blue) models in two representative locations of the shelf (a) and the deep basin (b), respectively. The location of the two profiles is identified by the red square (panel a) and triangle (panel b) in Fig. 2.

and Furner (2013) stretching formulation. In the case of GULF18-4.0, 35 ME $s$-levels are allocated to the upper sub-zone (i.e., between the free surface and the upper envelope) stretched according to Siddorn and Furner (2013) while 17 levels are used in the deeper part of the domain distributed to ensure that the vertical coordinate transformation and its Jacobian are continuous
(see Bruciaferri et al. (2018) for the details). Because of the Siddorn and Furner (2013) stretching formulation, the surface vertical level of GULF18-* configurations has a constant thickness of $1\ m$ while in PGM4 it ranges from $0.3\ m$ on-shelf to $6\ m$ off-shelf (Fig. 5). In addition, PGM4 presents uniformly distributed vertical levels in areas shallower than $150\ m$, while GULF18-* models switch off model levels' stretching only in areas shallower than $50\ m$ (Fig. 5a).

As shown in Fig. 3c and Fig. 5b, GULF18-4.0 is configured to have increased resolution in the proximity of the maximum depth of the upper envelope, which corresponds to the depth where the shelf-break occurs ($\approx 200\ m$). Such increased resolution near the envelopes is a feature of the ME system which helps to mitigate potential inaccuracies when simulating dense water cascading down a steep topography in those areas where model levels are not strictly terrain-following (e.g., see experiment two of Bruciaferri et al. (2018)).

**2.3  Dynamical core and model physics**

GULF18-4.0 is based on NEMO v4.0.4 (Madec and NEMO-team, 2019), GULF18-3.6 on NEMO v3.6-stable (Madec and NEMO-team, 2016) while PGM4 uses NEMO v3.4-stable (Madec and NEMO-team, 2012). NEMO v4.0.4 presents numerous differences from v3.4-stable, while v3.6-stable can be considered an intermediate release between the two - see http://forge.ipsl.jussieu.fr/nemo/wiki/Changelog for a comprehensive list of the main features of each NEMO release.

GULF18-* configurations build upon, and thus share, many of the core features of PGM4 model. For example, in order to accurately resolve the important tidal dynamics PGM4 and GULF18-* configurations implement a similar non-linear free surface via the NEMO variable volume layer (Levier et al., 2007) and time-splitting algorithms, using baroclinic and barotropic time-steps of $60\ s$ and $2\ s$, respectively. In addition, the three models also use the same pressure Jacobian scheme to compute

hydrostatic pressure gradients and the Energy and ENstrophy (EEN, Arakawa and Lamb (1981)) conserving scheme to advect momentum. Similarly, the $2^{nd}$ order Flux Corrected Transport (FCT) scheme, referred to as Total Variance Dissipation (TVD) scheme in the case of NEMO v3.4 (PGM4) or v3.6 (GULF18-3.6), is applied by all models to advect active tracers along with a non-linear equation of state based on EOS-80 formulation UNESCO (1983). The three models agree also on the turbulent bottom boundary layer formulation, implementing an implicit logarithmic bottom friction with a roughness length $z_0$

of $3 \times 10^{-3}\ m$ and a minimum drag coefficient $C_D$ of $2.5 \times 10^{-3}$. Finally, PGM4 and GULF18-* models compute the vertical eddy viscosity and diffusivity coefficients via the General Length Scale (GLS) turbulent closure scheme with identical settings apart from the minimum value of the turbulent kinetic energy $e$, which is $10^{-6}\ m^2\ s^{-2}$ in the case of PGM4 while is $10^{-7}$ $m^2\ s^{-2}$ for GULF18-* models.

GULF18-* and PMG4 configurations present also some important differences in the physics they implement (see Table 1). One major difference concerns the formulation of the lateral eddy fluxes. PGM4 is an eddy permitting model that needs to parameterize most of the mesoscale and the full sub-mesoscale eddy turbulence. Therefore it uses a Laplacian operator with a 3D constant diffusivity of $50\ m^2\ s^{-1}$ for tracers and a bi-harmonic operator with a constant viscosity of $-1 \times 10^{10}\ m^4\ s^{-1}$ for momentum. On the other hand, GULF18-* models are mesoscale eddy-rich configurations that only need to parameterise the

effect of unresolved eddies and the sub-mesoscale eddy activity. Both high resolution configurations use a horizontally aligned Laplacian operator for tracers and an along-levels oriented bi-harmonic operator for momentum but they differ in the formulation they adopt for the lateral mixing coefficients. GULF18-3.6 uses a modified version of the NEMO-3.6 code where the Smagorinsky formulation is extended also to tracers diffusion and therefore employs a Smagorinsky-like diffusivity ranging between $1\ m^2\ s^{-1}$ and $30\ m^2\ s^{-1}$. On the other hand, GULF18-4.0 uses the standard NEMO-4.0.4 code where such an option

is not available. Consequently, in the case of GULF18-4.0 it was preferred to test a 3D constant diffusivity of $2\ m^2\ s^{-1}$ (with correspondent eddy velocity and length scales $U_{scl} = 0.01\ m\ s^{-1}$ and $L_{scl} = 200\ m$, respectively) which could be used as a benchmark for future model developments. In the case of momentum, GULF18-3.6 applies a constant mixing coefficient of $-4 \times 10^8\ m^4\ s^{-1}$ while GULF18-4.0 uses a mesh size and depth dependent viscosity ranging between $3.84 \times 10^8\ m^4\ s^{-1}$ and $4.54 \times 10^8\ m^4\ s^{-1}$ (the correspondent velocity scale is $0.85\ m\ s^{-1}$).


In the case of PGM4, the NEMO code was modified to include a POLCOLMS-style scheme for the penetration of the incoming solar short-wave radiation and the model was set to use a fixed 1-D attenuation length scale of 6.49 m (Hyder et al., 2013). In contrast, GULF18-* configurations employ the standard NEMO RGB light penetration scheme where the penetration profile of the downward solar irradiance is function of various attenuation depth scales. For wavelengths longer than $700\ nm$,

a depth scale of $0.35\ m$ is applied. For shorter wavelengths, the visible light is split into three wavebands, blue (400-500

$nm$), green (500-600 $nm$) and red (600-700 $nm$) and for each waveband a different chlorophyll-dependent attenuation depth scale is used. In GULF18-* models, the fraction of short wave radiation that resides in the almost non-penetrative wavebands ($> 700$ $nm$) is set to NEMO default value of 58% while the chlorophyll concentration is set to NEMO default fixed value of 0.05 $mg\,m^{-3}$, corresponding to extinction depth scales of 2.62 $m$, 12.71 $m$ and 39.98 $m$ for red, green and blue wavebands, respectively.

## 2.4 External forcing and initialisation

Ocean simulations discussed in this manuscript are free-running (i.e., with no data assimilation) numerical experiments spanning five years, from 2013 to 2017. Each model is initialised from rest with temperature and salinity fields computed by PGM4 on 16 January 2013 (in the case of GULF18-* configurations a pre-processing 3D regridding procedure was applied ensuring that the water column was statically stable after the regridding ). The first year of the simulations is considered as spin-up time and hence is not included in the analysis.

At the surface, GULF18-* and PGM4 are forced with atmospheric fields from the Numerical Weather Prediction (NWP) configuration of the global Met Office Unified Model (Walters et al., 2019). NWP hindcast data are available at a horizontal resolution of $\approx 25$ $km$ before 2014-07-17, $\approx 17$ $km$ from 2014-07-17 to 2017-07-13 and $\approx 10$ $km$ after 2017-07-13. A major difference between GULF18-* and PGM4 configurations is the way they compute the boundary conditions at the surface. PGM4 uses directly prescribed NWP fluxes (i.e., the NEMO flux formulation) computed by the atmospheric model via the COARE4.0 algorithm (Walters et al., 2019) and including three-hourly data for heat and freshwater fluxes and hourly data for the momentum flux. On the other hand, GULF18-* models apply the Common Ocean-ice Reference Experiment (CORE) bulk formulae (Large and Yeager, 2009) to hourly data of wind speed at 10m and three hourly data of air temperature and specific humidity at 2m, short and long wave radiation and total precipitation from the NWP model to compute momentum, heat and freshwater fluxes at the air-sea interface.

At the single open boundary in the Gulf of Oman, GULF18-* and PGM4 apply a Flather (1976) radiation boundary condition to propagate tidal energy in the domain. In the case of GULF18-*, tidal elevation and velocity are derived from eight tidal constituents extracted from FES2014 gridded tidal analysis (Lyard et al., 2021) while PGM4 uses the TPXOv7.2 dataset (Egbert and Erofeeva, 2002). In this study, GULF18-* and PGM4 configurations are one-way nested within the Met Office Indian Ocean FOAM $1/12°$ model (Storkey et al., 2010). GULF18-* and PGM4 models use the flow relaxation scheme (Martinsen and Engedahl, 1987) to relax temperature and salinity fields to the values specified by the Indian Ocean FOAM $1/12°$ model over a ten point relaxation zone and the Flather boundary condition to add Sea Surface Height (SSH) and barotropic currents from the Indian Ocean FOAM system to the tidal constituents.

GULF18-* and PGM4 use climatological river run-off forcing. However, in PGM4 only the Shatt al-Arab (Tigris and Euphrates) river inflow at the Gulf's head is considered, while in GULF18-* domain also the Zohreh, Helleh, Mond and Minab rivers are included.

## 3   Models' evaluation approach

In this study we assess and compare the skills of PGM4, GULF18-3.6 and GULF18-4.0 models in reproducing the observed Gulf ocean dynamics during the period 2014-2017. Such a time-frame was chosen considering the number of available observations for the validation.

In addition to the hydrodynamics simulations, we conducted also Lagrangian experiments to assess the accuracy of the surface dynamics reproduced by our three Gulf models. This is a widely used methodology to validate and analyse the surface dynamics simulated by free running (e.g., Carniel et al. (2009); Dagestad and Röhrs (2019); Amemou et al. (2020); Paquin et al. (2020)) as well as assimilating (e.g., Barron et al. (2007); De Dominicis et al. (2016); Bruciaferri et al. (2021)) ocean models. Numerical experiments consisted of forcing a Lagrangian particle transport model with surface current velocities computed by PGM4 and GULF18-* models to numerically reproduce the trajectories of satellite-detected drifter tracks.

The next three Sections describe the observational datasets used for the verification (Sec. 3.1), the set-up of the additional Lagrangian simulations (Sec. 3.2) and the metrics used in the assessment (Sec. 3.3).

### 3.1   Observational datasets

Observations used to validate the numerical results include:

- Tidal constituents' amplitude and phase data computed by Pous et al. (2013) and Mashayekh Poul et al. (2016) conducting harmonic analysis on 34 tide-gauges recorded water-level time-series (see red triangles in Fig. 6 for location of the tide-gauges' included in the analysis).

- The Met Office Operational Sea surface Temperature and sea Ice Analysis (OSTIA) dataset (Donlon et al., 2012). This is a high resolution analysis of the global ocean sea surface temperature (SST) produced by combining satellite and in-situ SST observations with an accuracy (RMSE) of 0.57°C and zero bias (Donlon et al., 2012).

- The global ocean, near real-time (NRT), in situ quality controlled observational dataset (Wehde et al., 2021) from the Copernicus Marine Environment Monitoring Service (CMEMS). This dataset includes profiles of temperature (T) and salinity (S) from Conductivity-Temperature-Depth (CTD) measurements, T and S observations from ThermoSalinoGraphers (TSG) and satellite-tracked iSphere drifters trajectories. The locations of CTD and TSG measurements (squares and small circles, respectively) and where drifters were deployed (big circles) are shown in Fig. 6.

- Two hydrographic observational datasets. The first dataset includes 3 months measurements from mid-January to mid-April 2014 at a mooring station located approximately 44 km off the coast of Kuwait and 120 km south of the Gulf's

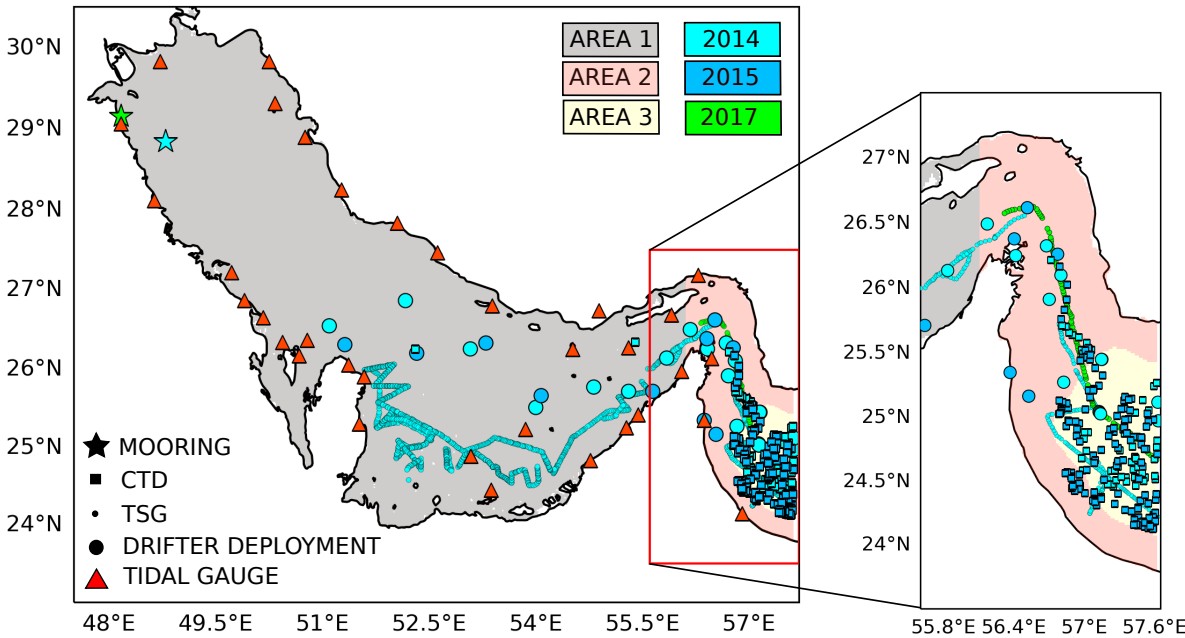

**Figure 6.** Map describing the three areas applied to analyse models' results together with the location and the temporal availability of the observations used in this study.

northern tip (see cyan star in Fig. 6 for location). The nominal water depth at mooring station was 23 m. The mooring was equipped with four high resolution temperature sensors (RBR, SOLO T) sampling at 2 Hz and two CTDs (RBR, XR-420) sampling every 18 seconds. Unfortunately, only the six instruments described above were recovered from the lower half of the mooring station and used for analysis. The second dataset includes 12 day measurements in July 2017 at mooring station located approximately 4 km off the coast of Kuwait (see green star in Fig. 6 for location). The nominal water depth at mooring station was 23 m. The mooring was equipped with nine high resolution temperature sensors (RBR, SOLO T) sampling up to 16 Hz and two CTDs (RBR, XR-420) sampling every 4 seconds. These two datasets are the only observations that provide time series of the water column vertical thermal structure in the northern Gulf, and several studies have analysed these data (e.g., Li et al. (2020); Al Senafi and Anis (2020b, a)).

Not all the observations are covering the entire period of the numerical experiments, see Table 2 and Fig. 6. The model validation has been tailored on the uneven distribution of the different type of observations.

### 3.2 Lagrangian simulations of iSphere drifters

iSphere drifters are half submerged spherical drifting buoys transported by surface ocean currents, wave-induced drift and the direct leeway of the wind (e.g., De Dominicis et al. (2016)). However, since the aim of our Lagrangian simulations was to validate ocean currents, in order to facilitate the results' interpretation it was preferred to not include the Stokes' drift in our

| MEASURED VARIABLE | INSTRUMENT | AVAILABLE OBSERVATIONS | | | | UNITS |
|---|---|---|---|---|---|---|
| | | 2014 | 2015 | 2016 | 2017 | |
| Water column T | CTD profiler | 61 | 136 | – | – | num. of downcasts |
| | Moored CTD | 80 | | | 14 | days of the timeseries |
| Water column S | CTD profiler | 55 | 136 | – | – | num. of downcasts |
| Near-surface T/S | TSG sampler | 1978 | – | – | 111 | num. of measurements |
| Lagrangian trajectories | iSphere drifters | 17 | 11 | – | – | num. of trajectories |

**Table 2.** Number and type of T/S and Lagrangian observations available for this study. The location of the measurements is shown in Fig. 6.

Lagrangian simulations, similarly to Barron et al. (2007); Carniel et al. (2009); Amemou et al. (2020).

We use OpenDrift Lagrangian framework (Dagestad et al., 2018; Dagestad and Röhrs, 2019) with a $4^{th}$ order Runge-Kutta scheme and a timestep of 3600 $s$ to integrate the following initial value problem for the drifter position $\boldsymbol{x}(t) = (x(t), y(t))$:

$$\begin{cases} \dfrac{d\boldsymbol{x}(t)}{dt} = \boldsymbol{u}(\boldsymbol{x}(t),t) + \boldsymbol{u}_w(\boldsymbol{x}(t),t) + \boldsymbol{u}'(\boldsymbol{x}(t),t) \,, & \text{(1a)} \\ \boldsymbol{x}(t_0) = \boldsymbol{x}_0 \,, & \text{(1b)} \end{cases}$$

where $\boldsymbol{x}_0$ is the initial drifter position at time $t_0$, $\boldsymbol{u}(\boldsymbol{x}(t),t)$ represents the surface Eulerian currents computed by the three Gulf models, $\boldsymbol{u}'(\boldsymbol{x}(t),t) = \alpha R$ with $R \in [-1,1]$ and randomly sampled from a uniform distribution and $\alpha = 0.04 \ m \ s^{-1}$ is used to simulate sub-grid turbulent diffusion and $\boldsymbol{u}_w(\boldsymbol{x}(t),t)$ is the wind drag velocity parameterised as $\boldsymbol{u}_w(\boldsymbol{x}(t),t) = \gamma \, \boldsymbol{U}_{10}(\boldsymbol{x}(t),t)$, with $\boldsymbol{U}_{10}(\boldsymbol{x}(t),t)$ the wind velocity at $10 \ m$ (from NWP fields) and $\gamma = 0.01$ in agreement with De Dominicis et al. (2016).

In order to maximise the usability of the observational dataset and to reduce the separation distance between observed and simulated track to an acceptable level (e.g., Dagestad and Röhrs (2019)), available satellite-tracked drifters trajectories were chunked into segments of 48 h duration. Then, similarly to Bruciaferri et al. (2021), for each segment 100 numerical drifters were released at the same initial location and time and the drift 48 h ahead was computed.

### 3.3 Evaluation metrics

In the case of tidal components and T/S measurements, the accuracy of PGM4 and GULF18-* models is quantified using the following metrics:

– Mean Bias Error:

$$MBE = N^{-1} \sum_{i=1}^{N} (x_{i,m} - x_{i,o}) \qquad (2)$$

– Root Mean Square Error:

$$RMSE = \left[ N^{-1} \sum_{i=1}^{N} (x_{i,m} - x_{i,o})^2 \right]^{1/2} \tag{3}$$

– Correlation coefficient:

$$r = \frac{\sum_{i=1}^{N} (x_{i,m} - \overline{x}_m)(x_{i,o} - \overline{x}_o)}{\left[ \sum_{i=1}^{N} (x_{i,m} - \overline{x}_m)^2 (x_{i,o} - \overline{x}_o)^2 \right]^{1/2}} \tag{4}$$

where $N$ is the total number of available observations, $x_{i,m}$ and $x_{i,o}$ are the values of the $i^{th}$ realisation of model and observational datasets, respectively, with mean values $\overline{x}_m = N^{-1} \sum_{i=1}^{N} x_{i,m}$ and $\overline{x}_o = N^{-1} \sum_{i=1}^{N} x_{i,o}$.

For T/S observations, metrics are computed bilinearly interpolating hourly model outputs on the geographical location of each T/S measurement. Then, in the case of hydrographic datasets, both observed and modelled profiles are also linearly interpolated on 26 reference depths with increased vertical resolution (from $2.5\ m$ to $25\ m$) in the first $200\ m$ of the water column. Harmonic analysis for computing models' tidal constituent amplitudes and phases is carried out using hourly Sea Surface Height (SSH) model fields for the year 2014. Then, the comparison with observations is conducted considering the closest grid point to the location of each tide-gauge.

The accuracy of the Lagrangian simulations is quantified using the Liu and Weisberg (2011) skill score ($ss$). This metric evaluates the separation between modelled and observed drifter trajectories normalized by their total length:

$$s = \frac{\sum_{i=0}^{N} d_i(\boldsymbol{x}_s(t_i), \boldsymbol{x}_o(t_i))}{\sum_{i=0}^{N} l_{oi}(\boldsymbol{x}_o(t_0), \boldsymbol{x}_o(t_i))}, \tag{5}$$

where $N$ is the total number of observed drifter positions in a given trajectory, $t_i$ is the time at which the $i^{th}$ drifter position has been recorded, $t_0$ is the time at which the drifter has been deployed, $d_i$ are distances between simulated $\boldsymbol{x}_s(t_i)$ and observed $\boldsymbol{x}_o(t_i)$ drifter positions at time $t_i$ and $l_{oi}$ is the length of the observed trajectory at time $t_i$. The skill score $ss$ is then defined as

$$ss = \begin{cases} 1 - s & , \text{if } s \leq 1, \\ 0 & , \text{if } s > 1, \end{cases}$$

so that $ss = 1$ indicates a perfect simulation while $ss = 0$ identifies a simulation with no skill. For each drifter simulation, 100 particles were released at the same initial location and time and the skill score of each numerical track was computed following Bruciaferri et al. (2021).

Considering the chaotic turbulent nature of the ocean dynamics and that our models are not taking advantage of data assimilation to constrain the predicted internal variability, it cannot be expected that our simulations accurately predict the space

and time location of small scale fronts and eddies. In addition, whilst increasing the resolution of an ocean model typically allows one to better resolve finer-scale features, metrics based on direct point matchup between interpolated model data and observations could not improve with higher granularity. This is due to the double penalty effect (e.g., Crocker et al. (2020)): features correctly predicted but misplaced with respect to the observations are penalised twice, for not occurring at the observed location and at the same time for occurring at the location where they were not observed. In this study, we found at least comparative performance of the high resolution GULF18 models with PGM4 using traditional verification techniques for the majority of the metrics included in the analysis. In the case of Lagrangian simulations, forcing the particle tracking model with surface currents affected by the double penalty effect will generate numerical trajectories that significantly differ from the observations, with a departure angle that could be as large as 180° and inevitably resulting in a poor average skill score. Therefore, two types of analysis were conducted to assess the surface currents: the first one considered all the available Lagrangian simulations (a total of 310 iSphere trajectories, 242 in 2014 and 68 in 2015) while the second one excluded from the analysis those trajectories that presented a skill score $ss < 0.35$ for all the three Gulf models (resulting in a total of 183 iSphere trajectories, 145 in 2014 and 38 in 2015). Such an approach should help us to investigate the impact of potential double penalty biases on our results.

Models' evaluation metrics are computed considering the entire basin or dividing the domain in three zones as shown in Fig. 6: the shelf area (longitude $< 56.1°E$, grey zone), the deep basin (depth $> 300\ m$, yellow area) and the shelf-break zone (longitude $> 56.1°E$ and depth $< 300\ m$, pink area).

## 4 Results and discussion

### 4.1 Tidal harmonics

The Gulf presents a complex tidal regime, characterised by tidal standing waves varying from being primarily semi-diurnal to diurnal and a large tidal range, with M2 peak amplitudes $> 1\ m$ throughout the whole domain (e.g. Proctor et al. (1994); Hyder et al. (2013)). The Gulf topography includes a shallow zone near the closed end which combines with an asymmetric cross-sectional depth profile (see Fig. 2). This particular conformation of the basin leads the generation of resonant interactions between semi-diurnal and diurnal waves resulting in tidal amplification at the northern end of the basin and a Kelvin-Taylor type system of amphidromic points shifted towards the coast to which the reflected Kelvin wave is bound (Roos and Schutte-laars, 2011). Consequently, semi-diurnal constituents present two amphidromic points in the northwestern and southern ends of the Gulf while diurnal constituents have a single amphidromic point in the central western part of the basin (Pous et al., 2013).

Figure 7 presents co-tidal charts of the principal diurnal (K1, top row) and semi-diurnal (M2, bottom row) components of FES2014 dataset (a,d) and PGM4 (b,e) and GULF18-4.0 (c,f) models (for clarity, here and in Fig. 8a,b only GULF18-4.0 results are shown, being the differences between the two GULF18 configurations almost negligible). In general, the three models

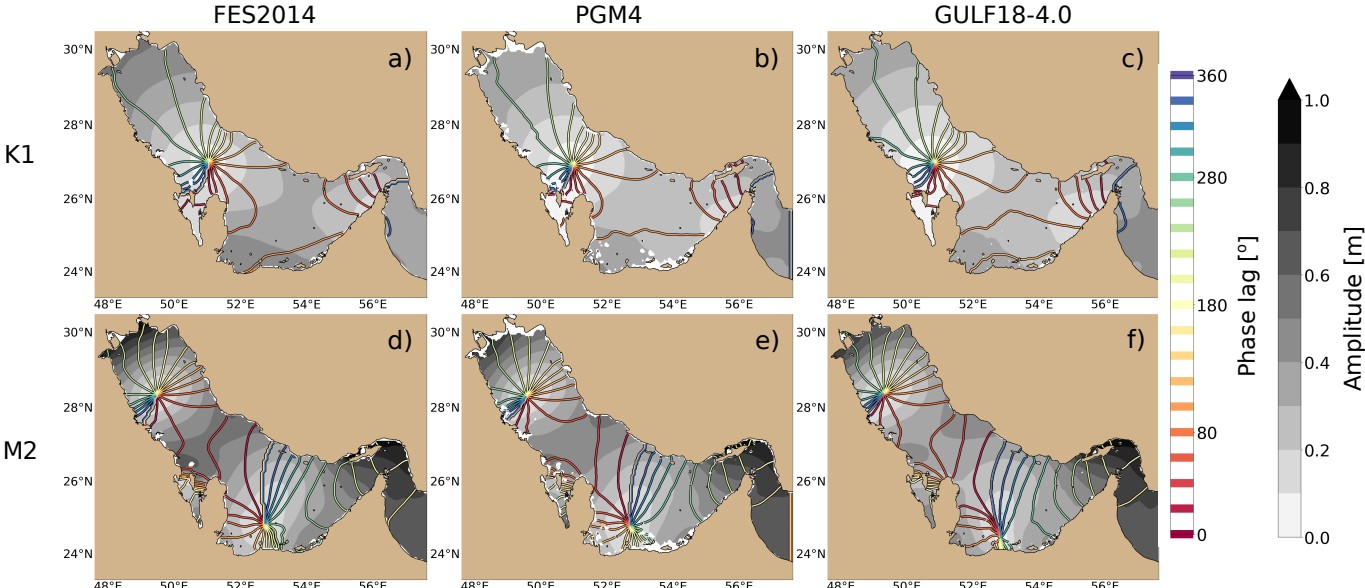

**Figure 7.** K1 (top panels) and M2 (bottom panels) co-tidal plots for FES2014 dataset (a,d), PGM4 (b,e), and GULF18-4.0 (c,f). Shading shows amplitude (m) while contours present the phase lag (°). GULF18-3.6 maps are not shown for clarity, being very similar to GULF18-4.0 results.

reproduce the typical pattern of the amphidromic points of the Gulf as in Pous et al. (2013). The main differences between PGM4 and GULF18-* models are where the coastline and the bathymetry differ the most, i.e. near the coasts of Qatar, Bahrain and south UAE.


The upper row of Fig. 8 presents the difference in the absolute errors of PGM4 and GULF18-4.0 models for M2 amplitude (a) and phase (b) for each tide-gauge included in the assessment. Similarly to Fig. 7, PGM4 has smaller errors than GULF18-* along the northwestern and western coast, especially in the proximity of Kuwait, Bahrain and Qatar. In those areas PGM4 bathymetry and land-sea mask were importantly modified (see Fig. 2) in order to improve the accuracy of the tidal dynamics represented by the model (Hyder et al., 2013). On the other hand, along the Iranian coast GULF18-* configurations seems

to have slightly improved accuracy. This is probably due to the better representation of the coastline in the higher resolution models which can affect the propagation of coastally-trapped Kelvin waves, especially in the case of near-resonantly forced Kelvin waves in channel-like basins (Griffiths, 2013).

The bottom row of Fig. 8 presents the RMSE against MBE of amplitude (c) and phase (d) computed by the models with respect to tide-gauges measurements for the seven tidal components included in the assessment. In general, the solutions of the three models for the phase lag are similar while for the amplitude, PGM4 seems to have a slightly better accuracy in the case of M2 and K1 components - e.g., in the case of M2 PGM4 presents MBE= $-2.6$ $cm$ and RMSE= $6.5$ $cm$, GULF18-3.6 has

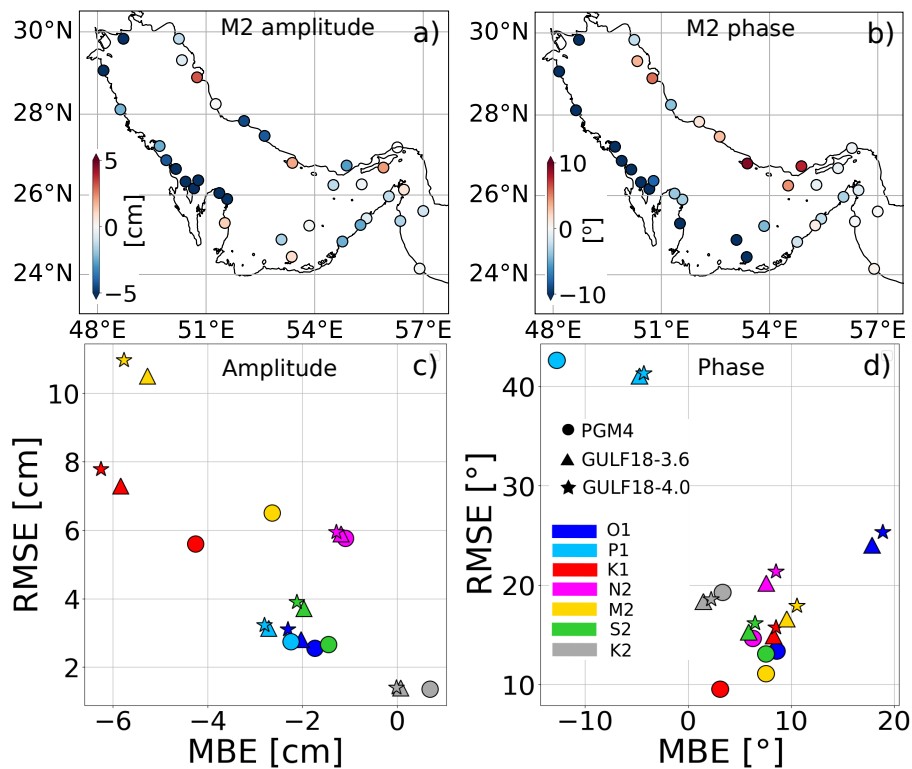

**Figure 8.** Upper row: difference in the absolute errors $|x_m - x_o|$ of PGM4 and GULF18-4.0 models for M2 amplitude (a) and phase (b) for each tide-gauge included in the assessment; Bottom row: Root Mean Square Error (RMSE) against Mean Bias Error (MBE) of amplitude (c) and phase (d) of the seven tidal components included in the assessment from harmonic analyses of model sea surface height and tide gauges.

MBE= $-5.3$ $cm$ and RMSE= $10.5$ $cm$ while GULF18-4.0 shows MBE= $-5.8$ $cm$ and RMSE= $11.0$ $cm$.


Shallow water waves propagate with a wave celerity proportional to $(gH)^{1/2}$, where $H$ is the water depth and $g$ the gravitational acceleration. Therefore, when applying the 'minimum depth approximation' as it is the case in our three models, the simulated tidal wave speed would be higher than the observed one. The total area where the three Gulf models apply the minimum depth approximation (see also Fig. 2d) is 33517 $km^2$ in the case of PGM4 while in GULF18-* is 39119 $km^2$. In

addition, in the proximity of the closed end of the domain where tidal waves are reflected and most of the resonant interactions occur, the area where the 10 $m$ approximation is applied is more extended in GULF18-* models than in PGM4 ($\approx 43\%$ larger). Hence, it is likely that the decreased accuracy of M2 and K1 amplitudes and phases in the north-western part of GULF18-* domain could be partly explained considering that in those areas GULF18-* models apply more often the minimum depth approximation in comparison to PGM4.

The small differences between GULF18-3.6 and GULF18-4.0 models (Fig. 8c and d) can be probably explained considering the additional smoothing of the upper envelope of GULF18-4.0 to reduce HPGE (see Sec. 2.2 for the details) and the different

value of the reference density used by the two models (see Tab. 1), similarly to O'Dea et al. (2017).

## 4.2 Sea surface temperature

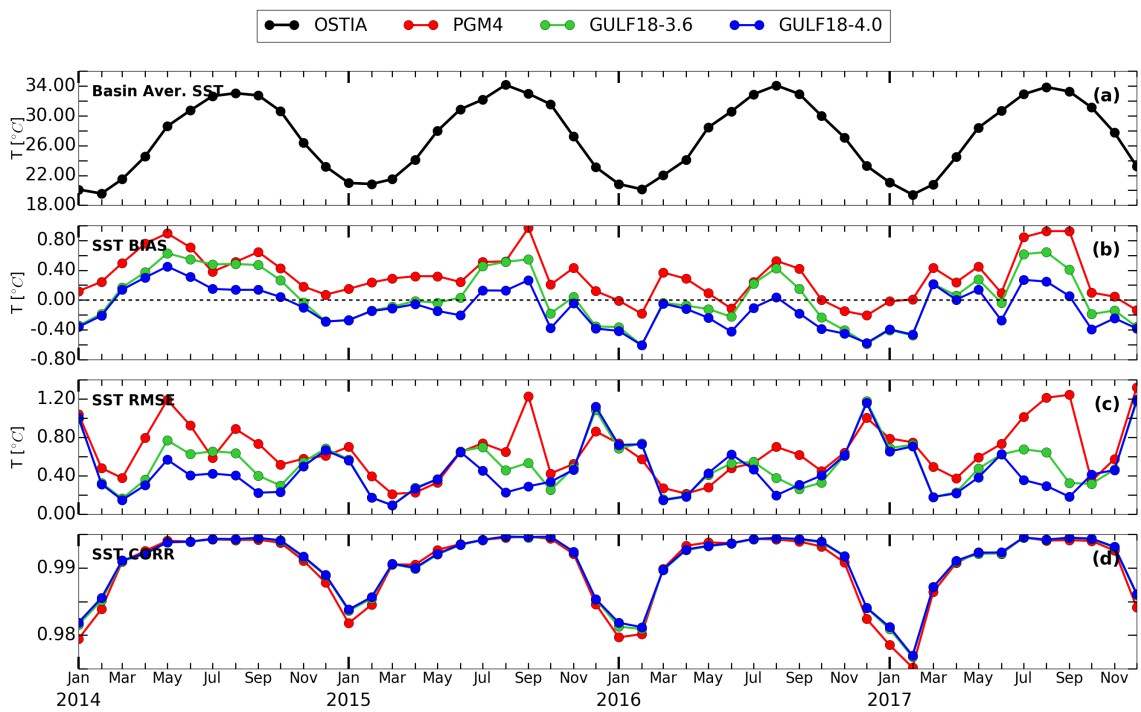

**Figure 9.** Monthly time-series of basin averaged OSTIA SST (a) and MBE (b), RMSE (c) and correlation coefficient $r$ (d) of PGM4 (in red), GULF18-3.6 (in green) and GULF18-4.0 (in blue) models.

SST strongly influences fluxes of heat, moisture and momentum across the ocean-atmosphere interface and the importance of more accurate SST simulations is widely recognised, both for long (e.g., Minobe et al. (2008)) and shorter (e.g., Mahmood et al. (2021)) timescales. In this section, we assess the skill of PGM4 and GULF18-* models in predicting the SST of the Gulf in the period 2014-2017.

Figure 9 presents monthly time-series of basin averaged MBE, RMSE and correlation coefficient $r$ of the three Gulf models with respect to OSTIA SST observed signal while Tab. 3 reports annual averages and standard deviations of models' MBE and RMSE. SST from model outputs was retrieved considering the temperature of the first model level. Numerical results demonstrate that the three models reproduce a seasonal and interannual variability in good agreement with OSTIA observations, with GULF18-* configurations having consistently improved accuracy in comparison to PGM4, both

in terms of MBE and RMSE. Mean metrics indicate that generally PGM4 is affected by a persistent warm bias (MBE=

| MODEL | 2014 | | 2015 | | 2016 | | 2017 | |
|---|---|---|---|---|---|---|---|---|
| | MBE | RMSE | MBE | RMSE | MBE | RMSE | MBE | RMSE |
| PGM4 | $0.45 \pm 0.25$ | $0.73 \pm 0.24$ | $0.36 \pm 0.22$ | $0.58 \pm 0.28$ | $0.11 \pm 0.24$ | $0.54 \pm 0.22$ | $0.33 \pm 0.37$ | $0.79 \pm 0.33$ |
| GULF18-3.6 | $0.21 \pm 0.33$ | $0.54 \pm 0.23$ | $0.04 \pm 0.29$ | $0.47 \pm 0.26$ | $-0.15 \pm 0.30$ | $0.50 \pm 0.27$ | $0.05 \pm 0.37$ | $0.54 \pm 0.26$ |
| GULF18-4.0 | $0.06 \pm 0.24$ | $0.43 \pm 0.23$ | $-0.20 \pm 0.19$ | $0.42 \pm 0.26$ | $-0.29 \pm 0.20$ | $0.50 \pm 0.28$ | $-0.10 \pm 0.27$ | $0.47 \pm 0.28$ |

**Table 3.** Annual means and standard deviations of MBE and RMSE of SST fields reproduced by PGM4, GULF18-3.6 and GULF18-4.0 models in comparison to OSTIA SST observations. Annual metrics for the correlation coefficient $r$ are not presented since the three models show very similar high values ($> 0.99$) during the whole simulated period.

$0.31 \pm 0.31°$C, RMSE$= 0.66 \pm 0.29°$C) while GULF18-3.6 presents a close to zero bias but larger RMSE and variability (MBE$= 0.04 \pm 0.38°$C, RMSE$= 0.51 \pm 0.26°$C) than GULF18-4.0, which has a slightly cold bias and the smallest RMSE and variability (MBE$= -0.11 \pm 0.26°$C, RMSE$= 0.46 \pm 0.26°$C).

Figure 10 explores the spatial distribution and the seasonal variability of models' errors. In general, GULF18-* configurations significantly reduce PGM4 inaccuracies throughout the domain and for all the seasons, with GULF18-4.0 presenting the largest improvements. In winter (DJF), GULF18-* models are able to mitigate the PGM4 warm bias in the central-eastern part of the domain and Strait of Hormuz and to reduce the marked cold bias of PGM4 along the western coast of the basin. In spring (MAM), both high resolution models present an overall small positive bias in contrast to the widely spread across the

domain SST overestimation of PGM4. This seems to be the case also in autumn (SON), although GULF18-4.0 seems to be affected by a slightly cold anomaly (especially in the central western part of the domain) that it is not present in GULF18-3.6. In summer (JJA), both GULF18-* configurations perform well, with GULF18-4.0 presenting the largest reduction of PGM4 errors, especially for the warm bias in the southern part of the domain and the cold anomaly in the proximity of the shallow closed end of the basin. In addition, both GULF18-* seem to introduce an SST underestimation in northern-central part of the

domain which appears to be more intense in the case of GULF18-4.0.

    SST biases often affect ocean models, particularly in summer when inaccuracies in the atmospheric forcings and/or in the upper mixed layer physics may be larger (e.g., Ezer and Mellor (2000), Hordoir et al. (2019) and Bruciaferri et al. (2020)) and SST data assimilation is typically used to constrain such model deficiencies (e.g, O'Dea et al. (2012); Hyder et al. (2013)).

In the case of our models, the improved accuracy of GULF18-* configurations in comparison to PGM4 could be due to differences in the horizontal resolution and sub-grid physics, the formulation of the surface boundary conditions, the light penetration schemes, the rivers forcing and the thickness of the first model level (as shown by Siddorn and Furner (2013)). One of the aims of this study was to assess the impact of the vertical coordinate system on the accuracy of a Gulf model. Therefore, a sensitivity test was conducted running GULF18-3.6 with a vertical coordinate system similar to the one of PGM4

(i.e., with an upper model level having a 2D varying thickness) to assess whether using a constant level thickness throughout the domain is important in terms of SST accuracy. Numerical results showed a basin averaged signal very similar to the orig-

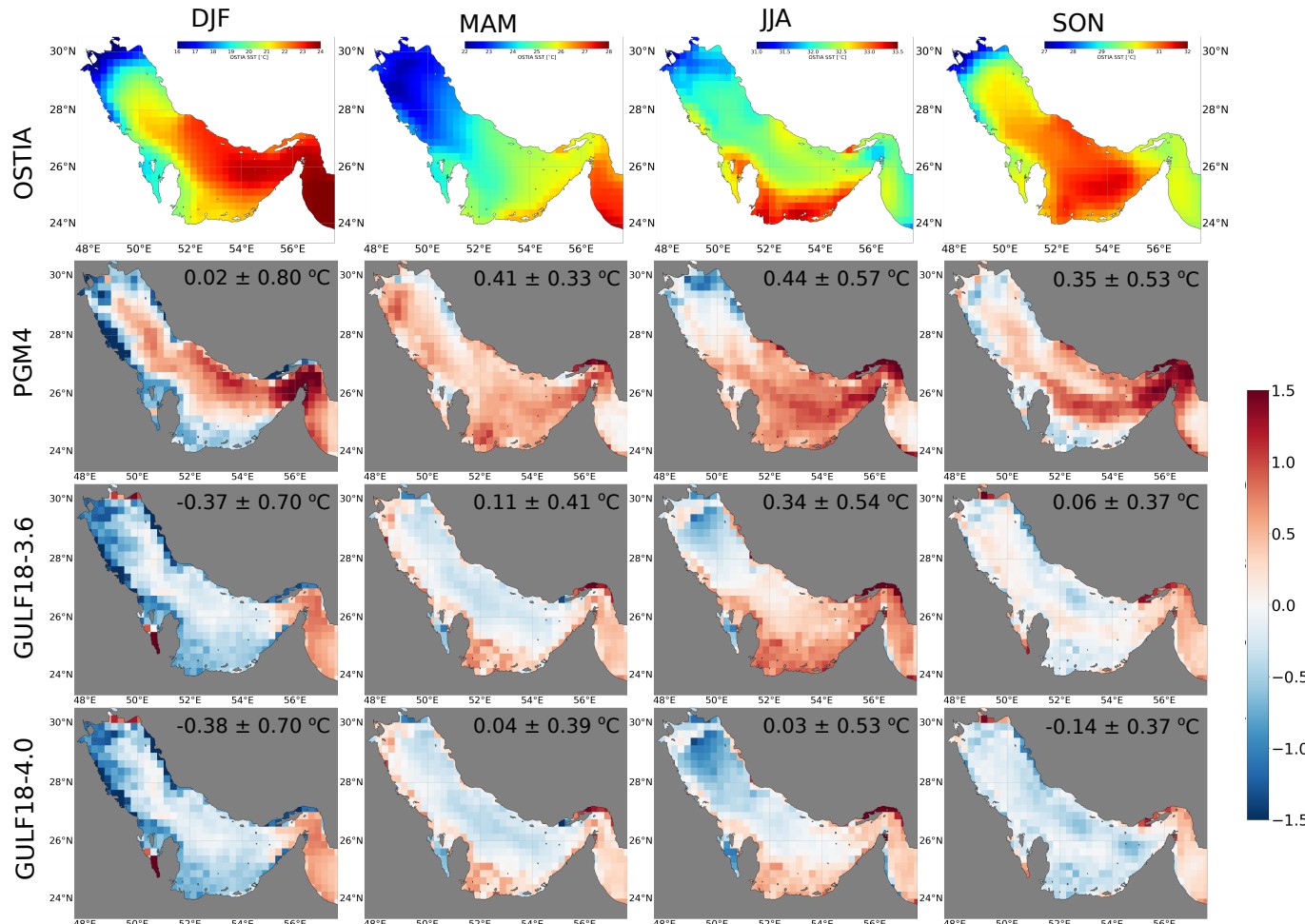

**Figure 10.** Seasonal OSTIA SST fields (top row) and seasonal SST anomalies (model minus observations) for PGM4 (second row), GULF18-3.6 (third row) and GULF18-4.0 (bottom row) models. All panels show 4-year mean anomalies for the period 2014-2017, with spatially averaged mean errors and standard deviations reported in each panel.

inal GULF18-3.6 simulation, suggesting a minor impact of the vertical coordinate system on the accuracy of the simulated SST.

GULF18-4.0 presents slightly improved accuracy in comparison to GULF18-3.6. This is consistent with the fact that the
two models differ only in the NEMO code revision, the vertical coordinate system below the sub-surface and some numerical/physical choices. Since the two GULF18 configurations present similar vertical resolution in the upper part of the water column, it is likely that the differences between the two models can be attributed to their different formulation of the diffusivity and viscosity (see Sec. 2.3 for the details).

## 4.3 Water column stratification

In this Section, the accuracy of PGM4 and GULF18-* models in reproducing the thermal and haline stratification of the Gulf during the period 2014-2017 is assessed against CTD measured T/S profiles, TSG sub-surface T/S observations and T time-series from two on-shelf moorings of the Kuwait coast.

Table 4 presents the average MBE and RMSE of the three Gulf models against CMEMS CTD measured T/S profiles for the three analysis areas defined in Sec. 3.3 and for the total domain. Overall, basin averaged metrics indicate that GULF18-4.0 has higher T accuracy when compared to PGM4 and GULF18-3.6, with larger improvements in 2014 than 2015, especially with respect to PGM4. To the contrary, in the case of S there does not seem to be a clear pattern - in 2014 GULF18-4.0 presents the highest accuracy while in 2015 it results to be the less precise, with differences in the basin averaged metrics of the models $< 0.03 \, PSU$.

**a) TEMPERATURE [$°C$]**

| MODEL | 2014 | | | | | | | | 2015 | | | | | | | |
|---|---|---|---|---|---|---|---|---|---|---|---|---|---|---|---|---|
| | AREA 1 (2) | | AREA 2 (12) | | AREA 3 (47) | | TOTAL (61) | | AREA 1 (0) | | AREA 2 (15) | | AREA 3 (121) | | TOTAL (136) | |
| | MBE | RMSE | MBE | RMSE | MBE | RMSE | MBE | RMSE | MBE | RMSE | MBE | RMSE | MBE | RMSE | MBE | RMSE |
| PGM4 | 1.48 | 2.81 | 0.64 | 0.80 | 0.50 | 1.14 | 0.56 | 1.12 | - | - | 1.30 | 1.60 | 0.60 | 1.06 | 0.68 | 1.12 |
| GULF18-3.6 | 0.70 | 1.80 | 0.39 | 0.60 | 0.67 | 1.28 | 0.62 | 1.16 | - | - | 0.90 | 1.34 | 0.79 | 1.20 | 0.80 | 1.22 |
| GULF18-4.0 | 0.59 | 2.10 | 0.22 | 0.53 | 0.51 | 1.13 | 0.46 | 1.05 | - | - | 0.72 | 1.25 | 0.66 | 1.11 | 0.66 | 1.12 |

**b) SALINITY [$PSU$]**

| MODEL | 2014 | | | | | | | | 2015 | | | | | | | |
|---|---|---|---|---|---|---|---|---|---|---|---|---|---|---|---|---|
| | AREA 1 (0) | | AREA 2 (11) | | AREA 3 (44) | | TOTAL (55) | | AREA 1 (0) | | AREA 2 (15) | | AREA 3 (121) | | TOTAL (136) | |
| | MBE | RMSE | MBE | RMSE | MBE | RMSE | MBE | RMSE | MBE | RMSE | MBE | RMSE | MBE | RMSE | MBE | RMSE |
| PGM4 | - | - | 0.10 | 0.16 | 0.10 | 0.23 | 0.10 | 0.21 | - | - | 0.20 | 0.47 | 0.09 | 0.22 | 0.10 | 0.25 |
| GULF18-3.6 | - | - | 0.03 | 0.10 | 0.10 | 0.23 | 0.09 | 0.20 | - | - | 0.08 | 0.42 | 0.09 | 0.22 | 0.09 | 0.24 |
| GULF18-4.0 | - | - | -0.02 | 0.09 | 0.08 | 0.21 | 0.06 | 0.18 | - | - | 0.05 | 0.43 | 0.10 | 0.25 | 0.09 | 0.27 |

**Table 4.** Mean MBE and RMSE of PGM4, GULF18-3.6 and GULF18-4.0 models when compared against CMEMS CTD T (a) and S (b) profiles for the three areas defined in Sec. 3.3 as well as for the whole domain in 2014 and 2015 (values between parenthesis indicate the number of observations included in the average).

Figure 11 presents T and S models' errors as a function of depth for the three areas considered in the analysis. In general, the three models seem to broadly overestimate T and S in the upper 200 $m$ of the water column, suggesting that the GLS turbulent closure scheme might need some tuning to improve the vertical mixing in the surface mixed layer.

On the shelf (Area 1), a limited number of available observations (two T profiles in 2014) seems to indicate that both GULF18-* configurations may have improved accuracy in comparison to PGM4. Panel 11a presents the on-shelf vertical dis-

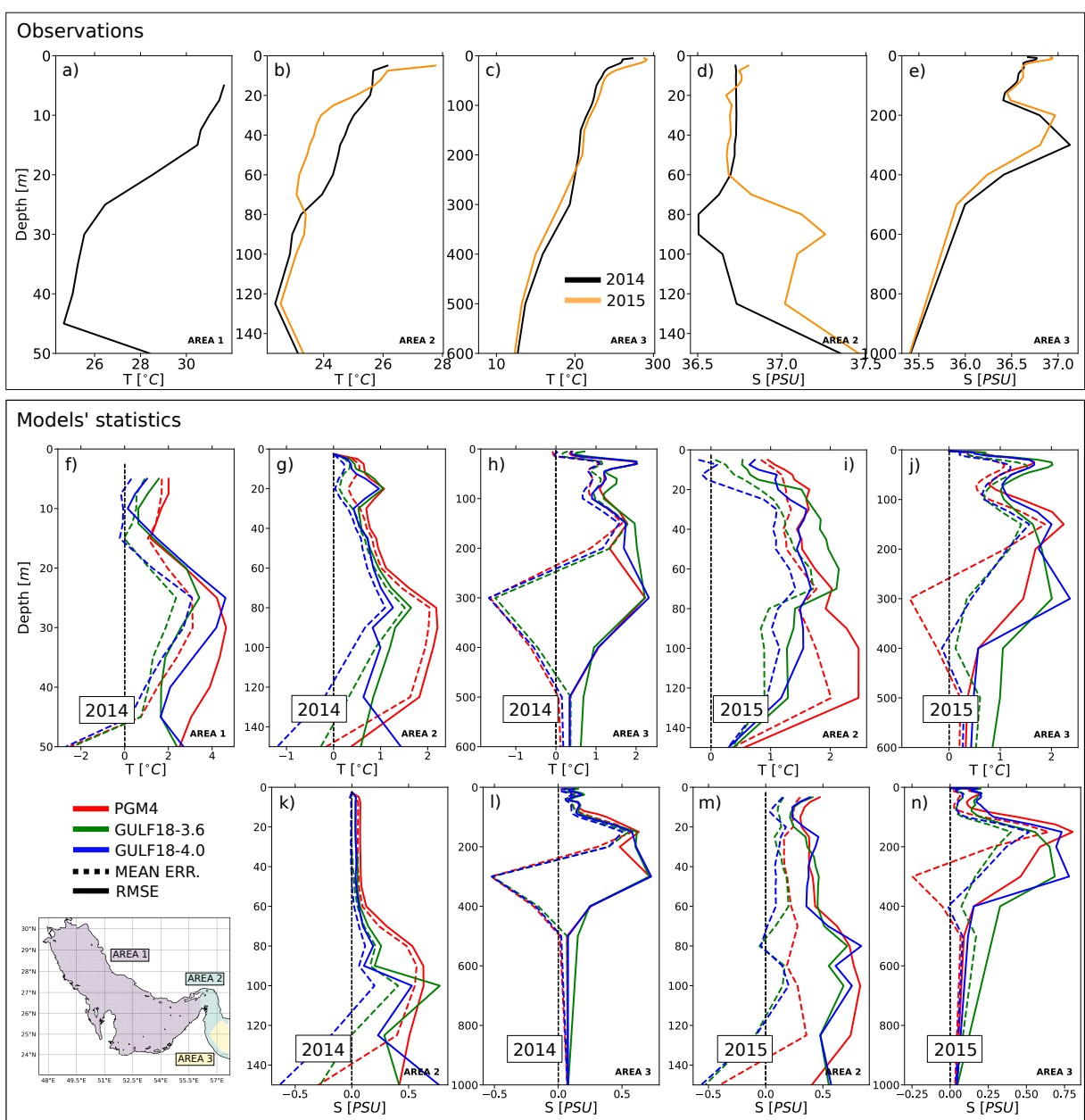

**Figure 11.** *Observations* panel: T (a,b,c) and S (d,e) vertical profiles averaged over the three analysis areas defined in Sec. 3.3 for 2014 (black) and 2015 (yellow); *Models' statistics* panel: vertical profiles of temperature (T, top row) and salinity (S, bottom row) MBE (dotted line) and RMSE (solid line) averaged over the three analysis areas for PGM4 (red), GULF18-3.6 (green) and GULF18-4.0 (blue) in 2014 and 2015. During the 2014-2015 period there were no available S observations for Area 1. Also, in 2015 there were no available T observations for Area 1.

tribution of T MBE and RMSE of the three models in 2014. Both GULF18-* configurations have improved accuracy in the proximity of the upper (0-10 $m$) and bottom (40-50 $m$) boundary layers with respect to PGM4, with GULF18-4.0 presenting slightly higher skill near the surface. This is probably due to the fact that in the upper and bottom mixed layers the two

GULF18-* models have increased vertical resolution in comparison to PGM4 (see Fig. 5). Similarly, the better performance of GULF18-3.6 at medium depths ($\approx 20 - 40\ m$) could be explained by its higher vertical resolution in this depth range with respect to the other two models.

Panels 11b, 11d and 11f, 11h show the vertical distribution of T and S models' errors, respectively, in the Strait of Hormuz

and near the shelf-break (Area 2). Generally, in this area GULF18-4.0 presents the smallest MBE and RMSE in the upper $\approx 100\ m$ for both T and S in 2014 and for T in 2015 while PGM4 has typically larger errors than GULF18-4.0 in the upper $\approx$ 120 $m$ of the water column for T in both years and for S in 2014. GULF18-3.6 has lower accuracy than GULF18-4.0 for T in the period 2014-2015 and for S in 2014, while there seems to be not a clear general pattern in comparison to PGM4. The higher skill of GULF18-4.0 in comparison to PGM4 can be probably explained by the lower vertical resolution of the latter model in

the upper 300 m of the water column in Area 2 and 3, as depicted in Fig. 5. In the case of the two GULF18-* configurations, while the higher vertical resolution of GULF18-4.0 at depths between 100 m and 200 m (see Fig. 5b) is likely to play a role, the lower accuracy of GULF18-3.6 can be probably partially attributed also to its larger inaccuracies in computing HPGs in the proximity of the shelf-break (see Fig. 4), in agreement with the findings of Wise et al. (2021) for a model of the European NWS.

In the deeper part of the domain (Area 3), GULF18-4.0 and PGM4 present, in general, a similar higher accuracy than GULF18-3.6 for T, while for S there is not a clear pattern (see Tab. 4). Figures 11c, 11e and 11g, 11i report the vertical distribution of models' MBE and RMSE in Area 3. In the upper $\approx 100\ m$ of the water column PGM4 seems to have a similar (or slightly better in 2015) skill than GULF18-4.0 for T, while GULF18-3.6 shows the lowest accuracy. For S, in the upper $\approx 200\ m$ of the water column GULF18-* models show consistently better accuracy than PGM4, with GULF18-3.6 showing

the best improvements, especially in 2015. Below $\approx 200\ m$, where the dynamics is typically more stagnant, PGM4 shows consistently better accuracy than the new GULF18-* configurations.

In Area 3, the dynamics of the three models is strongly influenced by the exchanges with the adjacent Indian Ocean. As explained in Sec. 2.4, all the three models apply a T/S relaxation zone of 10 grid points at the single lateral open boundary. However, given the coarser resolution of the 4 km model, this will result in a wider buffer zone in the case of PGM4, creating

T/S fields that are smoother and more heavily nudged to the data assimilating forcing at the open boundary. To the contrast, in the case of GULF18-* models the dynamics of Area 3 is less influenced by the open boundary and can evolve more freely. Therefore, whilst the good skill of the PGM4 here is partly due to the fact that a large portion of the deep area is strongly relaxed to the data assimilating solution forcing the open boundary, the higher skill of GULF18-4.0 for T and both GULF18-* for S in the upper part of the water column can be considered a model improvement over PGM4.


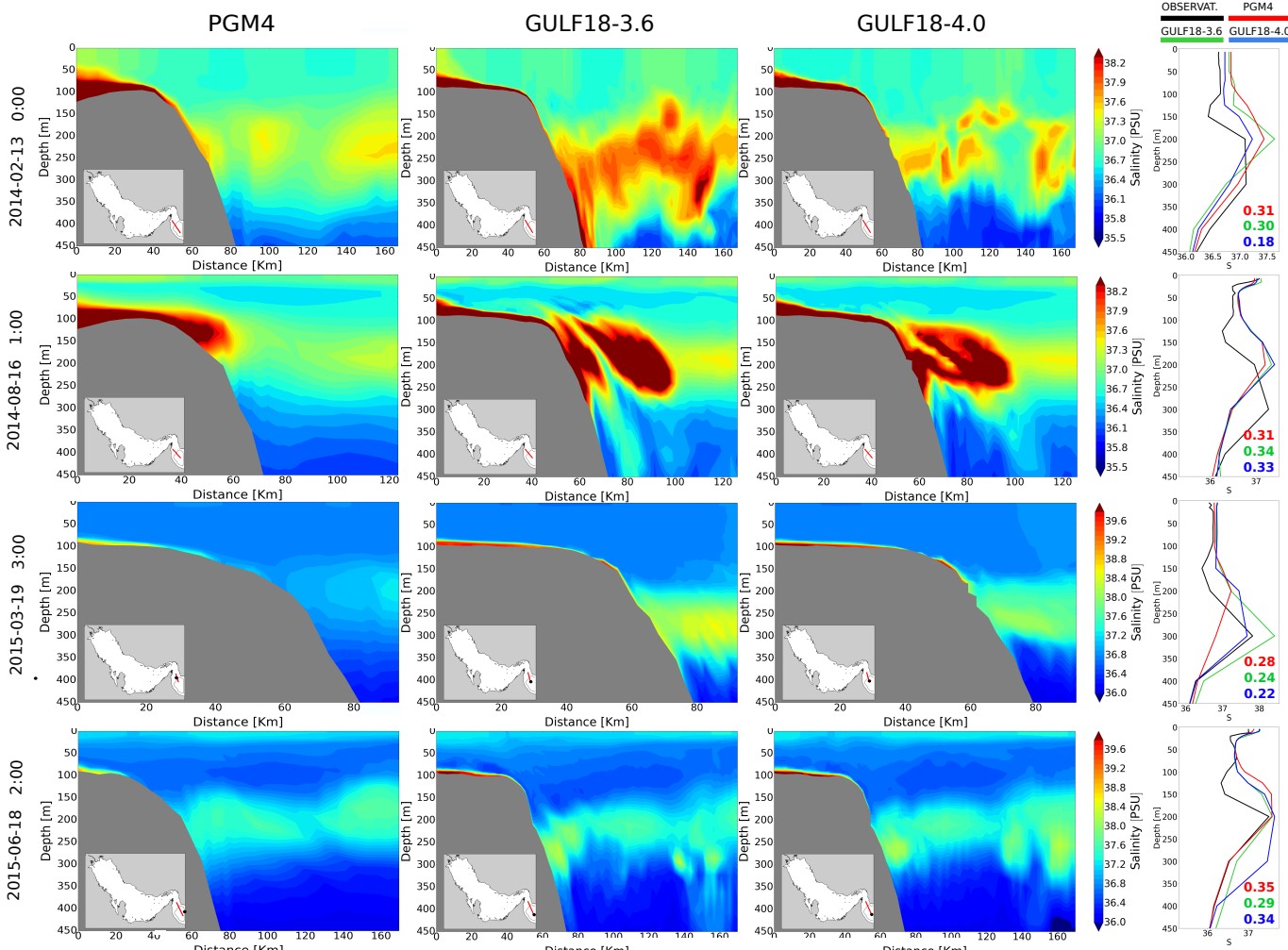

**Figure 12.** Examples of PGM4, GULF18-3.6 and GULF18-4.0 cross-sections representing salinity-driven cascading events occurred in 2014-2015. The location of the cross-sections is shown in the insets. The leftmost column of each row shows observed (in black) and modelled (PGM4 in red, GULF18-3.6 in green and GUF18-4.0 in blue) S profiles located at the off-shore end of each cross-section together with their RMSE.

In order to better understand the reasons behind the general improvements of GULF18-* configurations at the shelf-break (Area 2) and the upper part of the water column in Area 3, in Fig. 12 we investigate how the three models represented four salinity-driven cascading events observed in 2014-2015. Measured and modelled salinity profiles at the off-shore end of each cross-section are shown in the leftmost column of each row. In the case of GULF18-* models, gravity currents seems to be

affected by less numerical diffusion, enabling a stronger and more coherent cascading signal than in PGM4, where the solution appears to be generally smoother, more spread and less accurate in comparison to observations. The vertical grid of GULF18-4.0 has higher resolution in the proximity of the upper envelope (see Sec. 2.2). This enhanced vertical resolution seems to be

| a) TEMPERATURE $[°C]$ | | | | |
|---|---|---|---|---|
| **MODEL** | **2014** | | **2017** | |
| | MBE | RMSE | MBE | RMSE |
| PGM4 | 0.68 | 0.91 | 0.61 | 0.70 |
| GULF18-3.6 | 0.44 | 0.78 | 0.35 | 0.41 |
| GULF18-4.0 | 0.19 | 0.68 | 0.32 | 0.39 |
| b) SALINITY $[PSU]$ | | | | |
| **MODEL** | **2014** | | **2017** | |
| | MBE | RMSE | MBE | RMSE |
| PGM4 | 0.59 | 1.07 | -0.02 | 0.07 |
| GULF18-3.6 | 0.86 | 1.45 | -0.07 | 0.10 |
| GULF18-4.0 | 0.71 | 1.33 | -0.07 | 0.09 |

**Table 5.** Average MBE and RMSE of PGM4, GULF18-3.6 and GULF18-4.0 models when compared against CMEMS TSG T (top row) and S (bottom row) sub-surface hourly measurements collected in 2014 (from 23 to 25 March and from 21 May to 1 June) and 2017 (from 1 to 2 March). The location of the observations is shown in Fig. 6.

able to mitigate potential errors in the deeper part of the domain where model levels are not strictly terrain following, resulting in GULF18-* models having similar accuracy when simulating bottom intensified gravity currents.


Table 5 reports the average MBE and RMSE of the three models with respect to TSG sub-surface (between 0 and 5 $m$) hourly observations of T and S. In 2014, TSG measurements are located in the shallow southern part of the shelf (area 1) while in 2017 they are along a transect crossing Area 2 (see Fig. 6). For T, the assessment against TSG, SST and CTD observations seems to agree - GULF18-* configurations consistently present higher accuracy than PGM4, with GULF18-4.0 showing the larger
improvements. For S, limited TSG observations indicate a not clear pattern. In 2014, the three models seem to be affected by a sub-surface saline bias in the shallow southern part of the shelf, with GULF18-* models presenting larger error than PGM4. On the other hand, in 2017 the three models present similar small errors for the sub-surface S (differences in RMSE are $< 0.03$ as shown in Tab. 5).

We conclude the analysis of this Section by assessing how the models represented the evolution of the thermal stratification of the water column against two mooring temperature time-series collected in 2014 (Fig. 13) and 2017 (Fig. 14) off the coast of Kuwait (see Fig. 6 for the location). Unfortunately, in 2014 instruments attached to the upper part of the mooring failed to record and only bottom observations are available. In general, during January-April 2014 GULF18-3.6 shows an average MBE with magnitude of $\approx 0.2 - 0.4°C$, corresponding to a slightly cold bias with respect to observations, especially in the first $\approx$
20 days of the time-series (Fig. 13c and f). Conversely, in comparison to observations PGM4 presents a consistent warm bias for the first two months of the assessed period (Fig. 13b and e) , with larger errors than GULF18-3.6 (the average difference of the absolute value of GULF18-3.6 and PGM4 MBEs is $\approx -0.3/-0.6°C$). On the other hand, GULF18-4.0 presents a very

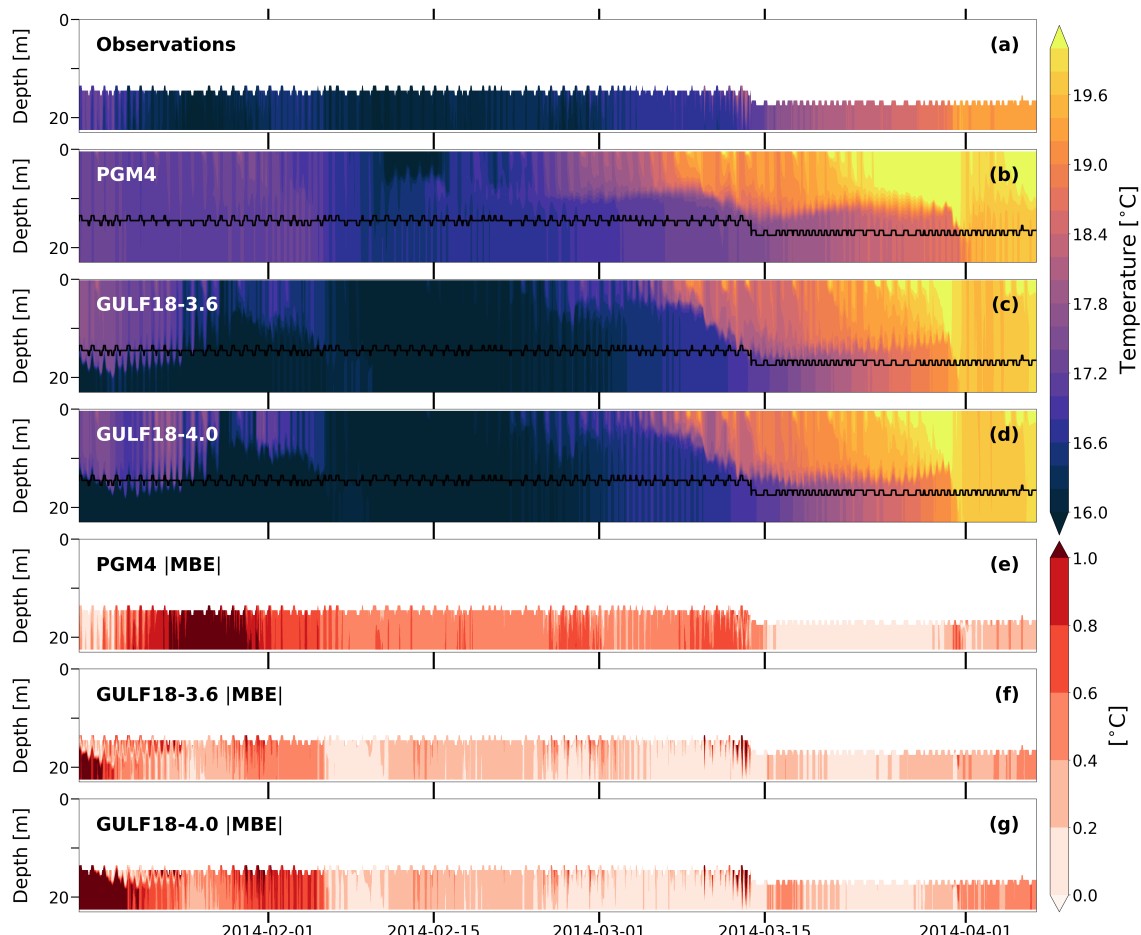

**Figure 13.** Hourly time-series of: a) mooring temperature profiles observed from 16 January to 6 April 2014 off the coast of Kuwait in the location identified by the cyan star in Fig. 6; temperature profiles computed by PGM4 (b), GULF18-3.6 (c) and GULF18-4.0 (d) and interpolated in the location of the mooring; absolute value of GULF18-3.6 MBE (e); differences between the magnitude of GULF18-3.6 MBE and the absolute value of PGM4 (f) and GULF18-4.0 (g) MBEs, respectively. Instruments attached to the upper part of the mooring failed to record and only bottom observations are available for this period.

similar solution to GULF18-3.6 (Fig. 13d and g) , with differences between the magnitude of their MBEs of $\approx \pm 0.3°C$.

In July 2017, all the three models present a cold bias in comparison to observations. In the case of GULF18-3.6, cold anomalies of $\approx 1°C$ mainly interest the upper part ($\approx 10 - 15\ m$) of the water column (Fig. 14c and e). The same occurs for GULF18-4.0, although with slightly colder values of $\approx 1.2°C$ on average (Fig. 14d and g). Conversely, PGM4 is affected by very strong and consistent cold biases larger than $2°C$, especially at depth (Fig. 14b and f). The analysis presented in Fig.

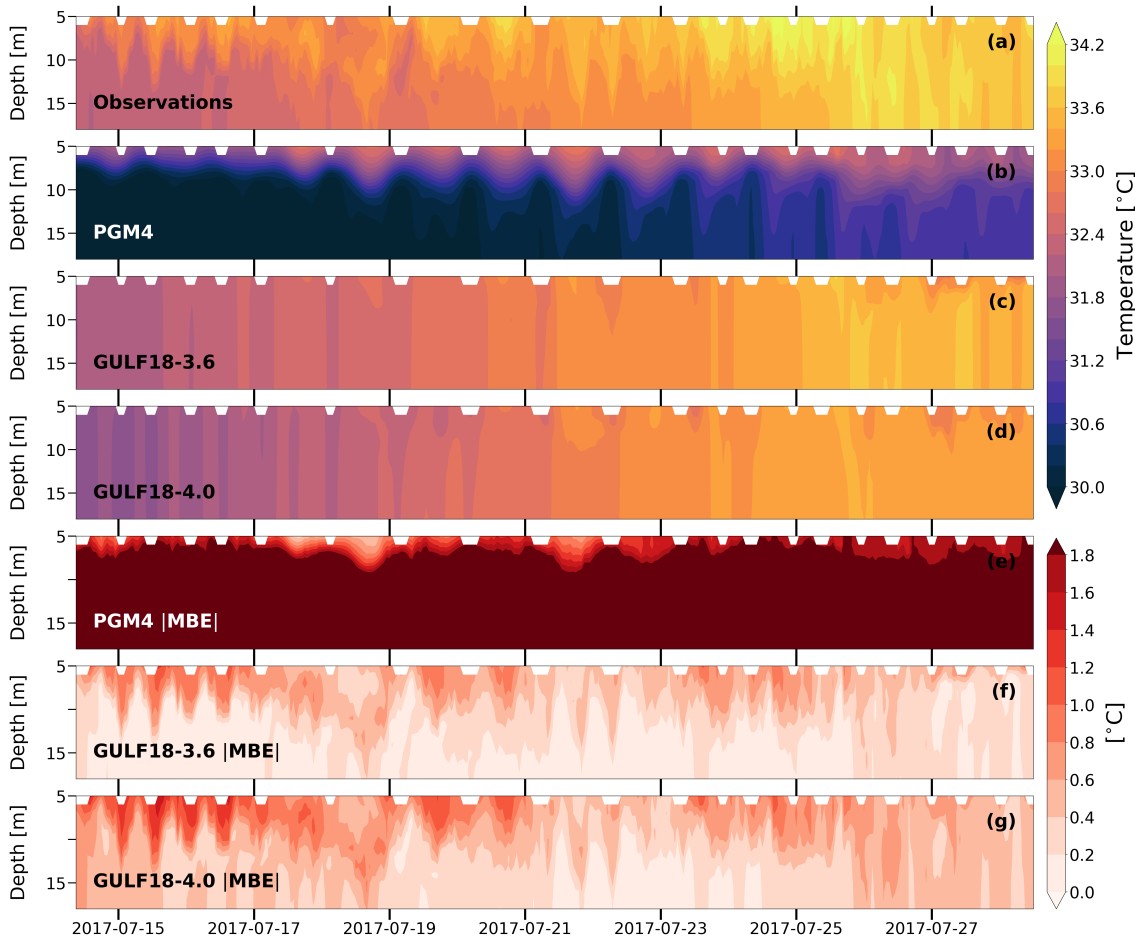

**Figure 14.** The same as in Fig. 13 but for the period from 14 to 28 July 2017.

10 for the seasonal variability of SST errors seems to agree well with the results shown here: PGM4 presents a cold bias in
summer (JJA) in the northern-west corner of the domain off Kuwait coasts that is not present in GULF18-* configurations.

### 4.4   Sea surface currents

One of the main purposes of an ocean forecasting system is to provide accurate data on the sea surface circulation to support,
for example, search and rescue or oil spill and plastic dispersal monitoring and control operations (e.g. Proctor et al. (1994);
Breivik et al. (2013)). In this Section we evaluate the skill of PGM4 and GULF18-* models in drift prediction against a number
of 48 h long observed drifters trajectories.

Table 6 presents the average skill score $ss$ and standard deviation of our Lagrangian simulations for the three areas defined
in Sec. 3 as well as for the whole domain during the period 2014-2015. Table 6a presents metrics computed considering all

a) ALL THE AVAILABLE NUMERICAL TRACKS

| MODEL | 2014 | | | | 2015 | | | |
|---|---|---|---|---|---|---|---|---|
| | AREA 1 (77) | AREA 2 (89) | AREA 3 (76) | ENTIRE DOMAIN | AREA 1 (46) | AREA 2 (22) | AREA 3 (0) | ENTIRE DOMAIN |
| PGM4 | $0.61 \pm 0.23$ | $0.57 \pm 0.20$ | $0.40 \pm 0.18$ | $0.53 \pm 0.22$ | $0.61 \pm 0.20$ | $0.42 \pm 0.15$ | – | $0.55 \pm 0.21$ |
| GULF18-3.6 | $0.64 \pm 0.20$ | $0.50 \pm 0.26$ | $0.40 \pm 0.21$ | $0.51 \pm 0.25$ | $0.56 \pm 0.22$ | $0.47 \pm 0.23$ | – | $0.53 \pm 0.23$ |
| GULF18-4.0 | $0.59 \pm 0.24$ | $0.50 \pm 0.25$ | $0.34 \pm 0.22$ | $0.48 \pm 0.26$ | $0.53 \pm 0.24$ | $0.41 \pm 0.19$ | – | $0.49 \pm 0.23$ |

b) ONLY NUMERICAL TRACKS WITH $ss \geq 0.35$

| MODEL | 2014 | | | | 2015 | | | |
|---|---|---|---|---|---|---|---|---|
| | AREA 1 (60) | AREA 2 (59) | AREA 3 (26) | ENTIRE DOMAIN | AREA 1 (32) | AREA 2 (6) | AREA 3 (0) | ENTIRE DOMAIN |
| PGM4 | $0.69 \pm 0.16$ | $0.66 \pm 0.12$ | $0.50 \pm 0.09$ | $0.65 \pm 0.15$ | $0.70 \pm 0.10$ | $0.56 \pm 0.11$ | – | $0.68 \pm 0.11$ |
| GULF18-3.6 | $0.71 \pm 0.12$ | $0.64 \pm 0.15$ | $0.57 \pm 0.13$ | $0.66 \pm 0.14$ | $0.67 \pm 0.12$ | $0.64 \pm 0.10$ | – | $0.67 \pm 0.12$ |
| GULF18-4.0 | $0.69 \pm 0.14$ | $0.64 \pm 0.14$ | $0.56 \pm 0.09$ | $0.65 \pm 0.14$ | $0.67 \pm 0.14$ | $0.64 \pm 0.10$ | – | $0.66 \pm 0.13$ |

**Table 6.** Average Skill Score $ss$ and standard deviation of numerical Lagrangian simulations for the year 2014 and 2015 computed considering all the available numerical tracks (a) or only those simulations with $ss \geq 0.35$ for all the three models. Metrics are computed for each of the three areas defined in Sec. 3 and shown in Fig. 6 and the whole domain. Values between parenthesis indicate the number of tracks included in the averages.

the available Lagrangian simulations while statistics presented in Tab. 6b were computed including only those trajectories with

$ss \geq 0.35$ for all the three models.

On the shelf (Area 1), the majority of the tracks considered in the analysis are located in the southern side of the central Gulf. Figure 15 shows that, in this area, most of the satellite-tracked drifters consistently drifted in a southerly direction, demonstrating a persistent surface southward transport. Table 6a indicates that in Area 1 the three Gulf models present averaged $ss > 0.5$

during the whole period 2014-2015, suggesting that where wind and tides are the predominant forcing the surface dynamics reproduced by the three model is typically accurate enough to obtain skilful Lagrangian particle tracking. Metrics computed including only Lagrangian simulations with $ss \geq 0.35$ (see Tab. 6b) seem to confirm that the three models are representing a generally comparable southward coastal circulation in the central part of the Gulf (differences are $\leq 0.03$) which transports the numerical drifting buoys in good agreement with the real ones ($ss > 0.65$).


In the proximity of the Strait of Hormuz and the shelf-break (Area 2), numerical results seem to not indicate a clear pattern. In 2014, Lagrangian simulations present an average $ss \geq 0.5$ for all the three models (see Tab. 6a), with PGM4 showing the best accuracy ($+7\%$). On the other hand, in 2015 all the models show an average $ss \leq 0.47$, with GULF18-3.6 being the one with the highest skill score ($+5/6\%$). When excluding from the analysis the numerical tracks with $ss < 0.35$, the average skill

score of all the three models increase to values $\geq 0.56$ (see Tab. 6b), with improvements larger for the two GULF18 models than for PGM4. This can be explained considering that in Area 2 PGM4 is eddy-permitting (see Fig. 1b) while GULF18-* models are eddy-resolving everywhere but in shallow areas along the Iranian coasts (see Fig. 1c) and therefore more suscepti-

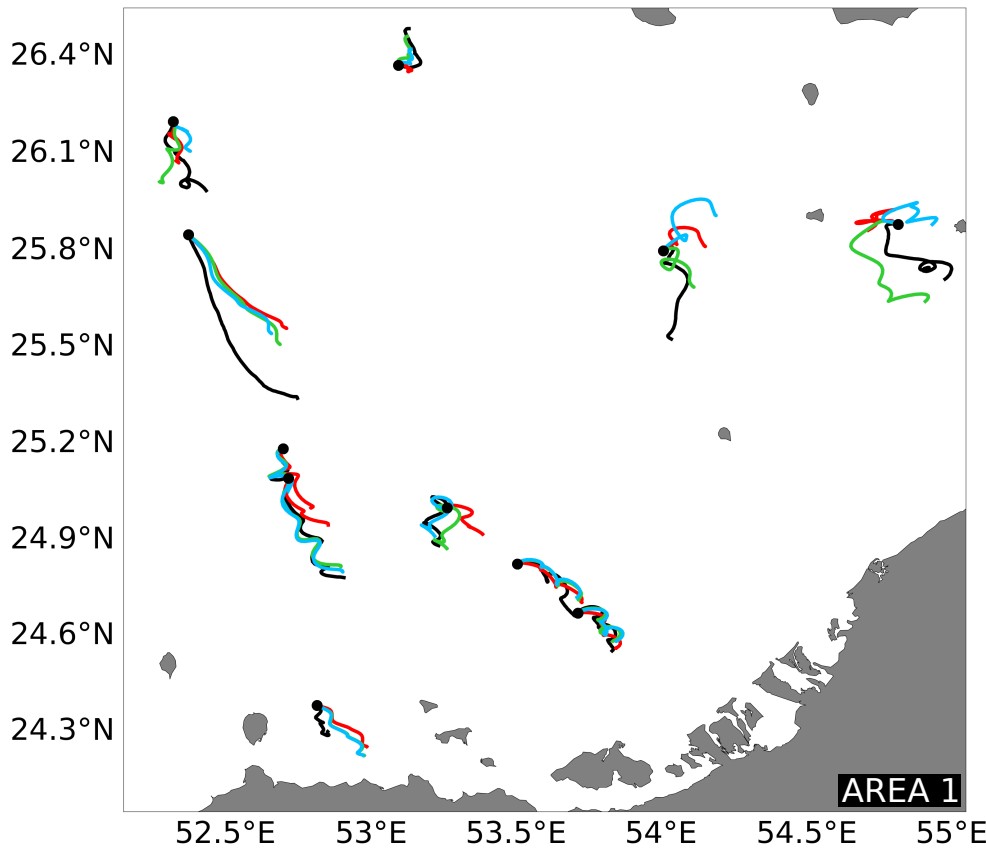

**Figure 15.** Examples of iSphere simulations located on the shelf (Area 1) during the 2014-2015 period. Observed drifter tracks are shown in black while numerical trajectories computed forcing the Lagrangian model with ocean surface currents from PGM4, GULF18-3.6 and GULF18-4.0 are shown in red, green and blue, respectively.

ble to the double penalty effect.

In the deep portion of the domain (Area 3), the three Gulf models consistently show a poor average $ss \leq 0.4$ when considering all the available drifter trajectories (see Tab. 6a). To the contrary, when including only simulations with $ss \geq 0.35$ the average skill score of the three models consistently improves to values $\geq 0.5$ (see Tab. 6b). In the open ocean, all the three models are eddy resolving (see Fig. 1) and the ocean dynamics is less controlled by tides. Therefore, these results might suggest that in this area double penalty biases could affect all the three ocean simulations. The visual inspection of entire

satellite-detected trajectories (see Fig. 16b for some examples) indicates the presence of a clockwise gyre in the western part of the Gulf of Oman, in agreement with existing literature (e.g. Reynolds (1993)) and the three models generally simulate consistent trajectories following such an anti-cyclonic circulation (Fig. 16b).

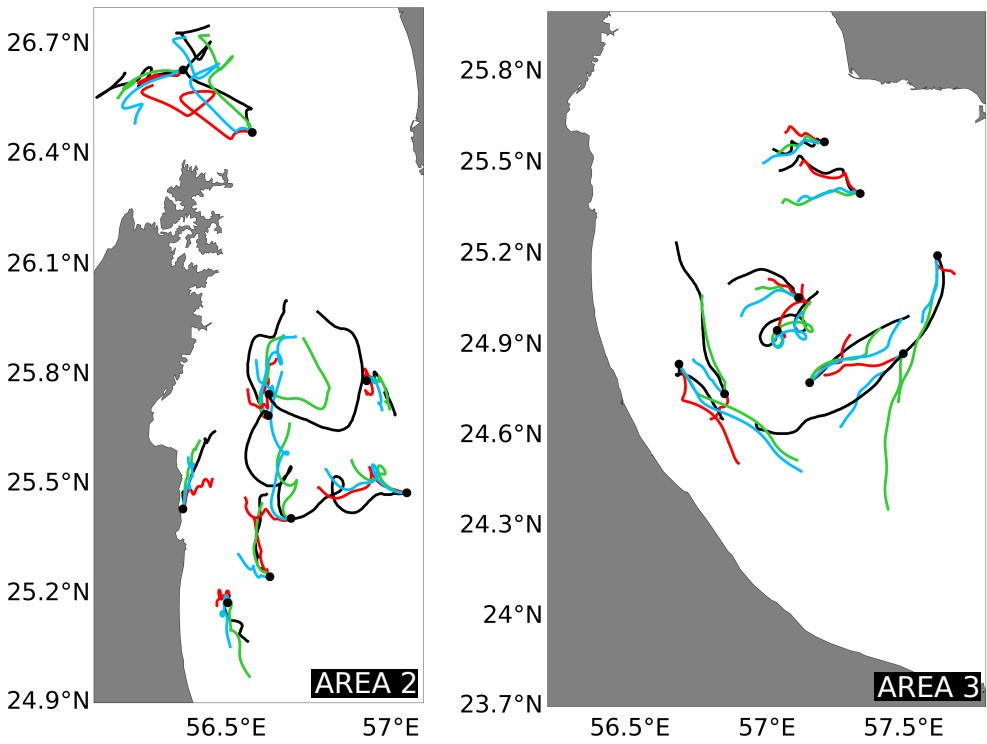

**Figure 16.** The same as Fig. 15 but for the Strait of Hormuz and shelf-break (Area 2, a) and the deep part of the domain (Area 3, b)

GULF18-4.0 appears to be the model that benefits the most from excluding skill-less simulations (i.e., $ss < 0.35$) from the analysis. On the one hand, this could be explained considering that PGM4 is eddy-resolving only in the deep part of the domain (Area 3, see Fig.1) and hence less prone to double penalty biases. On the other hand, at the surface the two GULF18-* models differ only for the NEMO code base and the lateral sub-grid parameterisations. Therefore, the highest accuracy of GULF18-3.6 surface currents is probably partly due to its larger values for the explicit diffusivity and viscosity (see Tab. 1 and Sec. 2.3 for the details) that are able to partially mitigate the negative impact of misplaced mesoscale structures.

Table 6b seems to indicate that, in general, GULF18-* models might present higher accuracy than PGM4 in Area 2 ($+8\%$ in 2015) and 3 ($+6/7\%$ in 2014) when excluding Lagrangian simulations with $ss < 0.35$ to remove possible double penalty biases. Figure 16a and b present examples of numerical and observed trajectories in Area 2 and Area 3, respectively. The visual inspection of the actual simulated tracks seems to indicate that in general PGM4 surface currents are slightly weaker than the real ones. Because of the Song and Haidvogel (1994) stretching function, in Area 2 and 3 PGM4 presents a surface layer thickness $> 5\ m$, while GULF18-* models, using a Siddorn and Furner (2013) stretching formulation, present a uniform $1\ m$ grid cell thickness at the surface (see Fig. 5). Hence, it is possible that part of the inaccuracies of PGM4 surface currents may be explained considering the too coarse resolution of the upper model layer, that may cause underestimation of the upper ocean shear and generates too weak grid cell averaged surface currents. Likewise, the larger lateral diffusivity of PGM4 may play

also a role in simulating smoother current fields. However, it is also possible that in the case of 2015 metrics under-sampling (6 tracks against 59 in 2014) may be affecting the results.

## 5  Conclusions and future work

The aim of this study was to investigate the impact of several science updates on the skills of a shelf sea model of the Gulf area and assess whether state-of-the-art ocean modelling practices and technologies were sufficient to improve its accuracy. In the specific, this work explored the sensitivity to changes in the bathymetry, lateral and vertical resolution, vertical coordinates and external forcing. Two high resolution (1.8 km, 52 vertical levels) Gulf models named GULF18-3.6 and GULF18-4.0 differing only in the vertical discretization scheme and the NEMO codebase (NEMO-v3.6 against NEMO-v4.0.4, respectively) have been developed and compared against the existing Met Office PGM4 model (4 km, 31 vertical levels, NEMO-v3.4). Both PGM4 and GULF18-3.6 use similar flavours of quasi-terrain-following vertical levels while GULF18-4.0 employs generalised multi-envelope (ME) vertical coordinates. The assessment compares non-assimilative hindcast integrations of the three Gulf models spanning the period 2014-2017 against available observations of the tidal dynamics, sea surface temperature, water column stratification and ocean currents at the surface.

Numerical results indicate that, overall, PGM4 and both GULF18 models give a comparable representation of the majority of the tidal constituents, despite their considerable differences in the domain geometry and tidal forcing. The three models use the same strategy of limiting the minimum depth of the model domain to deal with the large tidal excursion of the Gulf basin. Such a crude parameterization seems to be particularly penalizing in the case of the new high resolution models suggesting that, in order to get real benefit from using a more accurate and detailed bathymetry, the physical processes explicitly resolved by the model must be improved as well, in agreement with previous studies (e.g. Graham et al. (2018a); O'Dea et al. (2020)). Therefore, one future development will be the implementation of a wetting and drying algorithm to obtain a more realistic representation of the water level evolution.

GULF18-3.6 and GULF18-4.0 present similar skill for the tidal dynamics. This seems to indicate that using a ME vertical coordinate system optimised to reduce errors in the computation of the pressure force does not have significant detrimental impact on the accuracy of the simulated tides, in agreement with the findings of Wise et al. (2021).

Both GULF18 configurations present significantly reduced sea surface temperature (SST) biases in comparison to PGM4, improving the RMSE by $\approx 20\%$ in the case of GULF18-3.6 and $\approx 29\%$ for GULF18-4.0. While the increased resolution is likely to partially play a role in this, improvements are probably mainly due to processes that directly affect the local SST such as the surface fluxes formulation, river forcing and the light penetration scheme.

Although the two GULF18 models differ in the vertical coordinate system, they present similar resolution near the surface. Therefore, it is likely that the overall slightly better SST accuracy of GULF18-4.0 over GULF18-3.6 is due to the different sub-grid physics settings and the version of the NEMO code employed.

GULF18-4.0 seems to introduce a slightly cold bias in summer and autumn. Umlauf and Burchard (2005) showed the importance of carefully tuning the GLS vertical mixing scheme when dealing with stably stratified marine environment, as is the case for the shallow parts of the Gulf. Therefore, future work could involve sensitivity numerical tests to improve the vertical mixing at the surface.

Available observations of the water column thermal stratification indicate that both GULF18 configurations may have higher accuracy than PGM4 on the shelf, especially in the proximity of the upper (0-10 m) and bottom (40-50 m) boundary layers.

Similarly, in the proximity of the shelf-break both GULF18 models represent a more accurate vertical stratification than PGM4, with an average reduction of the RMSE of $\approx 22\%$ for temperature and $\approx 25\%$ for salinity. This is probably due to their more realistic bathymetry and enhanced vertical resolution in the upper 300 m of the water column, which allows the two GULF18 models to represent, for example, a more realistic saline dense water cascading at the shelf-break.

In the deep part of the domain, both high resolution configurations present a similar accuracy to PGM4 for temperature while for salinity there seems to be not a clear pattern. The good accuracy of PGM4 is probably due to the fact that, in this area, a larger portion of the model domain is strongly relaxed to the data assimilating solution forcing the open boundary. To the contrary, the improvements of GULF18 models are probably due to their higher horizontal and vertical resolution and their updated formulations of the atmospheric, river and light penetration forcings.

Our assessment seems also to suggest that, in general, GULF18-4.0 might have higher accuracy than GULF18-3.6 in representing the water column stratification, especially in the upper $\approx 120$ m near the shelf-break. This is probably due to the larger inaccuracies of GULF18-3.6 in computing the pressure gradient force in areas where the bottom topography is particularly steep, in agreement with the findings of Wise et al. (2021) for a model of the north-west European shelf.

Numerical trajectories simulated forcing a Lagrangian model with surface currents from PGM4 and GULF18 models were used to assess the accuracy of the simulated surface dynamics. On the shelf, where local wind and tides represent the leading dynamics, the numerical trajectories of all the three models are generally in good agreement with satellite-tracked drifters, with average $ss > 0.5$ and standard deviations $< 0.25$.

To the contrary, near the shelf-break and in the deep portion of the basin all the three models simulate Lagrangian tracks with a generally lower average skill score. In these areas, the barotropic tides are less important and the surface transport is mainly controlled by the wind- and buoyancy-driven circulation. Excluding from the analysis those numerical trajectories whose skill score is $< 0.35$ seems to suggest that double penalty biases could be one of the causes behind such a degradation of the surface dynamics. Therefore, in these areas data assimilation could be particularly useful to constrain models' drift and alleviate the negative impact of misplaced mesoscale structures.

Lagrangian experiments show that GULF18-* models are generally more prone to the double penalty effect. This is likely a consequence of the fact that PGM4 is an eddy-permitting model while GULF18-* models are eddy-rich configurations that resolve most of the mesoscale dynamics. However, our analysis seems also to indicate that double penalty biases might be particularly severe in the case of GULF18-4.0. This is probably related to the lower explicit diffusivity and viscosity adopted

by GULF18-4.0 and one future development will be trying different formulations for the lateral mixing coefficients of tracers and momentum.

In conclusion, our results indicate that both GULF18 models are broadly more accurate than the PGM4 model, proving the benefit of increasing the horizontal and vertical resolution. However, our tidal harmonic analysis suggests that future work may be needed in order to get real benefit from using a more realistic bottom topography as in the case of the GULF18 models. In addition, we found GULF18-4.0 to be generally more accurate than GULF18-3.6, demonstrating the advantage of optimising the vertical grid for the prevailing physical processes at stake, in agreement with previous numerical studies (e.g. Bruciaferri

et al. (2020); Wise et al. (2021)). The results of this study could be useful for the entire shelf/ocean modelling community, contributing to inform which new developments are needed to improve the physics represented by our ocean models.

In a future study, data assimilation could be applied to the GULF18-4.0 model to assess and understand the additional predictive skill that might be obtained on short-term forecasting time-scales. Similarly, GULF18-4.0 could be also used for

longer hindcast integrations to assess its skill on climatic time-scales and for future climatic projections of the Gulf marine environment.

*Code and data availability.*  The three Gulf models described and compared in this study are based on NEMO ocean model code, which is freely available from the NEMO website (www.nemo-ocean.eu). Additional modifications to the NEMO original code are required for running PGM4, GULF18-3.6 and GULF18-4.0 simulations. The actual NEMO source code, list of code branches, compilation keys and namelists adopted by the three models used in this manuscript are available at https://zenodo.org/record/6865886. Lagrangian sim-

ulations were run using OpenDrift Lagrangian modelling framework available at https://opendrift.github.io/. The nature of the 4-D data generated by the three models requires a large tape storage facility. The data that comprise the PGM4, GULF18-3.6 and GULF18-4.0 hindcast simulations are of the order of tens of TB. However, the data can be made available by contacting the authors. Processed data used in this paper for the production of figures and the analysis and the outputs of the Lagrangian simulations are available at https://zenodo.

org/record/6862364. OSTIA data are freely available via the European Copernicus Marine Environment Monitoring Service (CMEMS, https://marine.copernicus.eu/) at https://resources.marine.copernicus.eu/product-detail/SST_GLO_SST_L4_NRT_OBSERVATIONS_010_001/INFORMATION. Similarly, data for temperature and salinity of the Gulf water column and observed drifter trajectories are freely available at https://resources.marine.copernicus.eu/product-detail/INSITU_GLO_NRT_OBSERVATIONS_013_030/INFORMATION. Tidal observations from Pous et al. (2013) and Mashayekh Poul et al. (2016) are available at https://zenodo.org/record/6862364.

*Author contributions.*  DB developed GULF18-* models, ran GULF18-* simulations, ran the Lagrangian simulations, conducted the formal analysis, prepared the figures and wrote the draft of the paper. DB, MT and IA conceptualised the numerical experiments and gathered the available observations. IA ran PGM4 simulations. FA collected and provided the two hydrographic observational datasets off the coast of

Kuwait. EO provided the original code and support for the tidal harmonic analysis. All the co-authors contributed to the discussion of the results and to the final version of the manuscript.

*Competing interests.* The authors declare that they have no conflict of interest.

*Acknowledgements.* We thank two anonymous Reviewers for their thorough review of our manuscript and the constructive comments and suggestions that were made, they have greatly contributed to improving the manuscript. Simulations were carried out on the Cray HPC at the Met Office, UK. Funding support is gratefully acknowledged from the Ministry of Defense and the Public Weather Service. The projects collecting the northern Gulf datasets were funded by the Kuwait Foundation for the Advancement of Sciences, project numbers 2012640103
and P21644SE01.

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
