# Peer review of "GULF18, a high-resolution NEMO-based tidal ocean model of the Arabian/Persian Gulf"

_Geoscientific Model Development, 2022_

## Author Comment (AC1)

**Answers to comments of Reviewer 1**

This is the first review of « GULF18, a high-resolution NEMO-based tidal ocean model of the Arabian/Persian Gulf » by D. Bruciaferri et al. The study introduces two new dynamical ocean models of the Arabian/Persian Gulf. The authors evaluate in particular the effects of modifying the model vertical coordinates and the NEMO codebase against available observations of tides, hydrography and surface currents. They identify an improved representation of the Gulf hydrograpy and surface currents in the GULF18 models, with a slight added value of the GULF18-4.0 configuration. However, due to limitations in the representation of coastal shallow waters, tidal representation at the coast is degraded in the GULF18 models compared to the former PGM4 configuration. Overall, I found the manuscript very well written and illustrated. The authors have a deep insight into the NEMO model, ocean dynamics and the relevant physical processes at stake at the Gulf. They provide a thorough evaluation against a diversity of reference observations, despite the limited available in situ measurements in this basin. They provide overall thoughtful and convincing interpretations of the differences between models. I have one general concern but I would still recommend a minor revision given the quality of this work.

We thank the Reviewer for reading our Manuscript and giving very useful comments to improve its quality.

My main remark relates to the evaluation of simulated drifter tracks. The models being uninitialized, only the forced ocean dynamics (e.g. tides, Ekman currents) can be accurately reproduced, whereas internal variability ((sub)mesoscale eddies and filaments) suffers from the "double penalty effect", as mentioned by the authors. I feel that by removing trajectories with a skill score <0.35, the authors force a positive assessment on the ability of uninitialized runs to reproduce observed drifter tracks. I have listed below some suggestions to adapt this section in order to separate more explicitly the evaluation of the forced versus internal variability:

- what are the results when also including trajectories with ss<0.35? Are the conclusions unchanged regarding the comparison PGM4-GULF18 and the skill for those unassimilated runs?
- instead of a deterministic evaluation, would a statistical analysis (e.g. the surface geostrophic EKE against along-track altimetry, or the ensemble average dispersal rate across all trajectories) be more relevant to evaluate the internal dynamics of those unassimilated runs?
- given the fact that only the forced variability can be accurately represented by those unassimilated runs, wouldn't it be more relevant to evaluate specifically tidal and Ekman currents, instead of the total trajectories?

We thank the Reviewer for these comments, we will address them at point 25 of the "detailed comments" section below.

Detailed comments :

1. L.71-72: as shown in Table 1, the two GULF18 models do not only differ in the NEMO version and vertical discretization. I would moderate this statement.
   Thanks for the comment. We agree with the reviewer and the text is clarified at L73 of the new version of the paper.

2. Table 1: apart from model version and vertical coordinate, a large difference I see between both GULF18 versions is the value for the harmonic diffusivity. I have two questions with that regard:
   a) You mention a Smagorinsky-like diffusivity for GULF18-3.6, but in the NEMO 3.6 user book the Smagorinsky formulation concerns momentum viscosity, and not tracer diffusivity. Can you clarify that?
   b) What motivated you to switch to a constant value?

   GULF18-3.6 uses a modified version of the NEMO-3.6 code where the Smagorinsky formulation is extended also to tracers diffusion. On the other hand, GULF18-4.0 uses the standard NEMO-4.0.4 code where such an option is not available. Consequently, in the case of GULF18-4.0 it was preferred to test a 3D constant diffusivity which could be used as benchmark for future model developments. The text has been modified to clarify this point at L241-247.

3. L.104-105: could you calculate the Rossby radius from the model interannual mean density field to confirm that GULF18* models are eddy-resolving? Given the shallow bathymetry and the low stratification, I doubt that even at 1.8km resolution the first Rossby radius is resolved by a few (typically 5-10) model grid points.
   We thank the Reviewer for this comment. A map of the first baroclinic Rossby radius in the Gulf has been added (new Fig.1a). In addition, maps of the distribution of the metric $R_H = R_D/\Delta$ (where $\Delta = \sqrt{(\Delta x^2 + \Delta y^2)/2}$ and $\Delta x$ and $\Delta y$ are grid cells lateral sizes) proposed by Hallberg 2013 have been also added (new Fig. 1b and Fig. 1c) to show that PGM4 is an eddy-permitting model while GULF18-* are eddy-rich configurations. The text has been updated accordingly at L106-116 and L239-240.

4. L.162-169: you could mention where the mesh is terrain-following, and where it is geopotential, and relate that to the relevant processes to be resolved. For example, I note that the upper enveloppe is oriented along geopotential surfaces where H>180m, which I assume is relevant to minimize HPGE while accurately representing mixed layer processes.
   The suggestion has been included in the text at L177-180 and L182-184.

5. L.197: helps
   Done

6. L.198: from Fig.2c model levels seem to be terrain-following
   Agreed – we removed the reference to Fig.3c in the new version of the manuscript.

7. L.213: did you compare with the TEOS10 formulation, which has now been the reference EOS for seawater for over a decade?
   PGM4 and GULF18-* models use the EOS80 formulation as many other modern global and regional ocean models (e.g., Storkey et al. 2018, Graham et al. 2018, Wise et al. 2021).

8. L.237: did you check the occurrence of statistic instabilities upon initialization? Initialization in the winter increases the risk of such spurious initial behaviour due to the reduced stratification, which can degrade water mass properties.
   We thank the Reviewer for the comment. The initial condition has been checked for static instabilities after regridding in the case of GULF18-* models. Clarification has been made at L264-265.

9. L.245: for the PGM4 flux forcing, what Bulk formula was used in the forcing atmospheric model to produce the air-sea turbulent fluxes? Also, what is the forcing frequency?
   The global Met Office Unified Model used to compute the fluxes to force PGM4 uses the COAREv4 bulk formulae as specified in Walters et al., 2019. The text has been amended at L272-277 to clarify this point and to clarify the forcing frequency.

10. L.306: is R sampled randomly from a uniform distribution?
    Yes, it is – clarification has been made in the text at L339.

11. Equation 2: a parenthesis is missing to include the observed values
    Done, thanks.

12. L.349-352: even in the cases where ss>0.35, there might be some double penalty effect in the GULF18 simulations compared to PGM4. What about only comparing current statistics instead of simulating trajectories for those uninitialized runs?
    We thank the Reviewer for this comment. Drifter trajectories were the only available observations that we could use to have an estimate of the accuracy of the surface currents simulated by our Gulf models. Computing current statistics as suggested by the Reviewer will inevitably involve estimate pseudo-Eulerian current velocities from the Lagrangian drifter trajectories, a procedure which has its own difficulties and inaccuracies (see e.g., Blockley et al. 2012). Moreover, it will still require interpolating models' velocities at the location where the pseudo-Eulerian velocities are estimated (e.g., Blockley et al. 2012), and so being still potentially affected by the double penalty effect. In addition, the methodology we are using is a widely used technique to validate and analyse the surface dynamics simulated by free-running (e.g., Carniel et al. 2009, Dagestad et al. 2019, Amemou et al. 2020, Paquin et al.

2020) as well assimilating (e.g., Barron et al. 2007, De Dominicis et al., 2016, Bruciaferri et al. 2021) ocean models. The text has been clarified at L296-298.

13. Fig.6: could you add a reference cotidal chart from say the FES2014 reference data?
We thank the Reviewer for the suggestion – Fig.6 has been amended accordingly.

14. L.377-379: if the PGM4 land-sea mask and bathymetry modifications improve tidal representation at the coast, why not apply the exact same procedure to the GULF18 models?
We thank the Reviewer for this interesting question. While the PGM4 land-sea mask and bathymetry modifications improve tidal representation at the coast, they are based on model tuning and not on scientific justifications. To the contrary, the aim of this study was to investigate the impact of several science updates on the skills of a model of the Gulf dynamics and assess whether state-of-the-art ocean modelling practises and technologies were sufficient to improve its accuracy. Eventually, we think that the results of this study will be useful for the entire shelf / ocean modelling community, contributing to inform which new developments are needed to improve the physics represented by our ocean models. Some text on this point has been included in the Conclusions at L633-636 and L706-707 of the new version of the manuscript.

15. L.394: it is not clear to me from Fig.1d that the 10m bathymetry approximation is more frequent in GULF18 than in PGM4. It rather seems to apply to different locations.
We understand the Reviewer doubt and we try to clarify it by computing the total area where PGM4 and GULF18-* models apply the 10m approximation. In PGM4 the area is 33517 km2 while in GULF18-* models it is 39119 km2. We also compute the area where PGM4 and GULF18-* models apply the 10m approximation only for the north-west region, which is where tidal waves are reflected and most of the resonant interactions occur. In this region, the area affected by the 10m approximation is 3956 km2 in PGM4 while it is 5659 km2 in GULF18-* models. The text has been clarified at L434-437.

16. L.410: what is the depth difference between the first model level depth and the OSTIA reference depth used to reconstruct the observed SST? Can this difference affect the model evaluation, especially in the summer when near-surface temperature stratification is large?
The OSTIA system produces a foundation SST estimate, which is the SST free of diurnal variability (Donlon et al. 2012). As such, by definition there is not a reference depth for OSTIA (see also the explicative diagram at https://www.ghrsst.org/ghrsst-data-services/products/). On the other hand, ocean simulations like the ones from PGM4 and GULF18-* include diurnal cycle in their upper model level temperature. However, the comparison of modelled SST against OSTIA products is a widely used

verification practise (e.g. O'Dea et al. 2012, Zhang et al. 2016, Tonani et al. 2019, Bruciaferri et al. 2020).

17. L.413: regarding the PGM4 warm anomaly with respect to GULF18 models: are the parameters of the GLS scheme identical to the GULF18 models? What background vertical tracer diffusivity is used, and does it differ from the GULF18 models?
The setting of the GLS scheme is identical in all the three models apart from the minimum value of the turbulent kinetic energy, which is $10^{-6}$ m2/s2 in PGM4 and $10^{-7}$ m2/s2 in GULF18-*. The three models use the same value of $10^{-7}$ m2/s for the background vertical tracer diffusivity. The text has been clarified at L230-233 of the new version of the manuscript.

18. Regarding the solar radiation penetration, are the characteristic decay depths of the GULF18 models from the NEMO RGB scheme higher than that of PGM4?
For the penetrative solar radiation, PGM4 uses POLCOMS fixed length scale formulation with a characteristic attenuation depth scale of 6.49 m. On the other hand, GULF18-* models use the standard NEMO RGB formulation where the penetration profile of the downward solar irradiance is function of various attenuation depth scales. For wavelengths longer than 700 nm, a depth scale of 0.35 m is applied. For shorter wavelengths, the visible light is split into three wavebands, blue (400-500 nm), green (500-600 nm) and red (600-700 nm) and for each waveband a different chlorophyll-dependent attenuation depth scale is used.

In GULF18-* models, the fraction of short-wave radiation that resides in the almost non-penetrative wavebands (>700nm) is set to NEMO default value of 58% and the chlorophyll concentration is set to NEMO default fixed value of 0.05 mg/m3, corresponding to extinction depth scales of 2.62 m, 12.71 m and 39.98 m for red, green and blue wavebands, respectively. Figure 1 shows the penetration profile of the downward solar irradiance for PGM4 (in blue) and GULF18-* (in red) models. The text has been clarified at L253-260.

[Figure]

Figure 1 - Penetration profile of the downward solar irradiance for PGM4 (in blue) and GULF18-* (in red) models.

19. Fig.9-10: displaying reference observations before showing model biases would help identify the relevant patterns and water masses.
Done, thank you.

20. L.504: could you quantify that by performing the RMSE for those four vertical profiles?
Added RMSE for the three models in Fig. 12 as suggested.

21. L.513: there seems to be an overall sub-surface saline bias in the GULF18 runs, more so than for PGM4. did you compare the terms of the freshwater budget (evaporation, precipitation, river runoff)?
Thanks for the comment. During the model development, we did compare the freshwater budget of PGM4 and GULF18-* models to check that they were consistent – and that was actually the case. However, a more in detail analysis of the terms of the freshwater budget was not carried out and unfortunately, the three terms were not saved in our output files. However, we do not agree with the Reviewer's comment in this case. The sub-surface saline bias seems to be indicated only by limited TSG observations in 2014 while in 2017 the three models seem to be quite consistent (differences in RMSE are < 0.03 as shown in Tab 5). In addition, the analysis of the S vertical profile indicates that in the upper 20-30 m of the water column GULF18-* models are consistently more accurate or at least as accurate as PGM4 (see Fig. 11 of the new version of the manuscript).  At L556-559 of the new version of the paper the text has been clarified.

22. L.517-518: if Fig. 12 is not commented anywhere in the manuscript, I recommend removing it.

*Fig.12 was commented in the original manuscript, but the reference to the figure number was wrong, sorry for the confusion. The text has been amended accordingly between L564 and L570.*

23. Fig.13: I think it would read better with the bias of each model against observations for Fig.13e-g, and in the same order as the absolute values are shown (PGM4 and then both GULF18 runs)
*Thanks for the suggestion, Fig.12 and Fig.13 have been amended accordingly.*

24. L.532: accurate
*Amended, thanks.*

25. Table 6: those large skill scores give the impression that data assimilation is not necessary to predict particle dispersals, however all trajectories with ss<0.35 have been priorly removed. Some suggestions:
    a) what are the results when also including trajectories with ss<0.35? Are the conclusions unchanged regarding the comparison PGM4-GULF18 and the skill for those unassimilated runs?
    b) instead of a deterministic evaluation, would a statistical analysis (e.g. the surface geostrophic EKE against along-track altimetry, or the ensemble average dispersal rate across all trajectories) be more relevant to evaluate the internal dynamics of those unassimilated runs?
    c) given the fact that only the forced variability can be accurately represented by those unassimilated runs, wouldn't it be more relevant to evaluate specifically tidal and Ekman currents, instead of the total trajectories?

*We thank the reviewer for these comments.*

*We start replying to point b) and c). Using along-track altimetry as suggested at point b) could be quite challenging in the Gulf area, given the scarce availability of freely accessible altimetry observations and given the small dimensions of the basin.  Also, specifically evaluating tidal and Ekman currents as suggested at point c) would be very interesting and valuable but unfortunately not an option in our case, considering that drifter trajectories were the only available observations we could use to assess the accuracy of the surface dynamics simulated by our Gulf models. Finally, we would like to note that the method we are using is a widely used technique to validate and analyse the surface dynamics simulated by free running (e.g., Carniel et al. 2009, Dagestad et al. 2019, Amemou et al. 2020, Paquin et al. 2020) as well as assimilating (e.g., Barron et al. 2007, De Dominicis et al., 2016, Bruciaferri et al. 2021) ocean models.*

*We do understand the concern expressed by the Reviewer in the introduction and point a). For this reason, we include in Tab. 6 of the new version of the manuscript also the average ss computed including all the available drifter trajectories (Tab 6a).*

In light of the new results, the text has been amended at L377-379, L386-393, most of Sec. 4.4 and L684-699 .

26. L.546 and Fig.14: I do not see the cyclonic flow. Southward drift demonstrates an overall surface southward advection. The cyclonic flow would be an extrapolation assuming a northward return flow further east and an eastward boundary current along the southern boundary, which is not evident from the drifter trajectories.
We agree with the Reviewer's comment. The text has been amended at L589-591.

27. L.551: seem
Done
28. L.554: seems
Done
29. L.563: present
Done
30. L.628: generates
Done
31. L.653, 655: ran
Done

32. L.10-11: after reading the manuscript and in particular Fig.7, I feel that the sentence on comparable results across models for tides is not faithful to your results. I recommend to acknowledge in the abstract that tidal dynamics are degraded near the coast in the GULF18 simulations, e.g.: "For the surface currents the three models give comparable results. However, for the tides, the PGM4 overperforms the GULF18 models. Our tidal harmonics analysis suggests that..."
We agree with the Reviewer and the text in the abstract is amended at L11-13.

**REFERENCES**

Amemou, H., Koné, V., Aman, A., & Lett, C. (2020). Assessment of a Lagrangian model using trajectories of oceanographic drifters and fishing devices in the Tropical Atlantic Ocean. Progress in Oceanography, 188, 102. https://doi.org/10.1016/j.pocean.2020.102426

Barron, C. N., Smedstad, L. F., Dastugue, J. M., & Smedstad, O. M. (2007). Evaluation of ocean models using observed and simulated drifter trajectories: Impact of sea surface height on synthetic profiles for data assimilation. Journal of Geophysical Research, 112(C7), C07. https://doi.org/10.1029/2006JC003982

Blockley, E. W., Martin, M. J., and Hyder, P.: Validation of FOAM near-surface ocean current forecasts using Lagrangian drifting buoys, Ocean Sci., 8, 551–565, https://doi.org/10.5194/os-8-551-2012, 2012.

Bruciaferri, D., Shapiro, G., Stanichny, S., Zatsepin, A., Ezer, T., Wobus, F., Francis, X., and Hilton, D.: The development of a 3D computational mesh to improve the representation of dynamic processes: The Black Sea test case, Ocean Modelling, 146, 101 534, https://doi.org/10.1016/j.ocemod.2019.101534, 2020.

Bruciaferri, D., Tonani, M., Lewis, H.W., Siddorn, J. R., Saulter, A., Castillo Sanchez, J. M., et al. (2021). The impact of ocean-wave coupling on the upper ocean circulation during storm events. Journal of Geophysical Research: Oceans, 126, e2021JC017343. https://doi.org/10.1029/2021JC017343

Carniel, S., Warner, J. C., Chiggiato, J., & Sclavo, M. (2009). Investigating the impact of surface wave breaking on modeling the trajectories of drifters in the northern Adriatic Sea during a wind-storm event. Ocean Modelling, 30(2–3), 225–239. https://doi.org/10.1016/j.ocemod.2009.07.001

Dagestad, K.-F., & Röhrs, J. (2019). Prediction of ocean surface trajectories using satellite derived vs. modeled ocean currents. Remote Sensing of Environment, 223, 130–142. https://doi.org/10.1016/j.rse.2019.01.001

De Dominicis, M., Bruciaferri, D., Gerin, R., Pinardi, N., Poulain, P. M., Garreau, P., et al. (2016). A multi-model assessment of the impact of currents, waves and wind in modelling surface drifters and oil spill. Deep-Sea Research Part II: Topical Studies in Oceanography. 21–38. https://doi.org/10.1016/j.dsr2.2016.04.002

Graham, J. A., O'Dea, E., Holt, J., Polton, J., Hewitt, H. T., Furner, R., Guihou, K., Brereton, A., Arnold, A., Wakelin, S., Castillo Sanchez, J. M., and Mayorga Adame, C. G.: AMM15: a new high-resolution NEMO configuration for operational simulation of the European north-west shelf, Geoscientific Model Development, 11, 681–696, https://doi.org/10.5194/gmd-11-681-2018, 2018.

Hallberg, R.: Using a resolution function to regulate parameterizations of oceanic mesoscale eddy effects, Ocean Modelling, 72, 92–103, https://doi.org/10.1016/j.ocemod.2013.08.007, 2013

O'Dea, E. J., Arnold, A. K., Edwards, K. P., Furner, R., Hyder, P., Martin, M. J., Siddorn, J. R., Storkey, D., While, J., Holt, J. T., and Liu, H.: An operational ocean forecast system incorporating NEMO and SST data assimilation for the tidally driven European North-West shelf, Journal of Operational Oceanography, 5, 3–17, https://doi.org/10.1080/1755876X.2012.1102012, 2012.

Paquin, JP., Lu, Y., Taylor, S. et al. High-resolution modelling of a coastal harbour in the presence of strong tides and significant river runoff. Ocean Dynamics 70, 365–385 (2020). https://doi.org/10.1007/s10236-019-01334-7

Storkey, D., Blaker, A. T., Mathiot, P., Megann, A., Aksenov, Y., Blockley, E. W., Calvert, D., Graham, T., Hewitt, H. T., Hyder, P., Kuhlbrodt, T., Rae, J. G. L., and Sinha, B.: UK Global Ocean GO6 and GO7: a traceable hierarchy of model resolutions, Geoscientific Model Development, 11, 3187–3213, https://doi.org/10.5194/gmd-11-3187-2018, 2018.

Tonani, M., Sykes, P., King, R. R., McConnell, N., Péquignet, A.-C., O'Dea, E., Graham, J. A., Polton, J., and Siddorn, J.: The impact of a new high-resolution ocean model on the Met Office North-West European Shelf forecasting system, Ocean Science, 15, 1133–1158, https://doi.org/10.5194/os-15-1133-2019, 2019.

Wise, A., Harle, J., Bruciaferri, D., O'Dea, E., and Polton, J.: The effect of vertical coordinates on the accuracy of a shelf sea model, Ocean Modelling, p. 101935, https://doi.org/10.1016/j.ocemod.2021.101935, 2021.

Zhang, Y. J., Stanev, E. V., Grashorn, S. 2016. Unstructured-grid model for the North Sea and Baltic Sea: Validation against observations, Ocean Modelling, Volume 97, 2016, https://doi.org/10.1016/j.ocemod.2015.11.009.

---

## Author Response (AR2)

**Answers to comments of Reviewer 1 – second round**

This is the second review of « GULF18, a high-resolution NEMO-based tidal ocean model of the Arabian/Persian Gulf » by D. Bruciaferri et al. I am very satisfied with the comprehensive response and manuscript review provided by the authors. I recommend a minor revision.

We thank the Reviewer for the new comments.

I only have the following minor suggestions:

- In relation to my comments 9 and 18: I believe that the substantial differences between PGM4 and GULF18 in the formulation of the atmospheric forcing (namely, turbulent air-sea fluxes and solar radiation) should be mentioned when introducing both model configurations (e.g. in L.3-4 of the abstract and in L.73 of the introduction).
  We thank the Reviewer for this comment. We modify the text at L58-59 as suggested. However, in the case of the Abstract we prefer to change "external forcing" with "river and atmospheric forcing" at L2 and not adding more details since we think it will just add complexity which is not needed at this stage.

- In relation to my comments 9, 16, 17 and 18: I think that these elements can help interpret model biases and differences in the representation of SST (Fig.9-10 and Table 3). Specifically:

  - The OSTIA SST is a lower estimate of the actual daily average SST, so that typically half the diurnal cycle should be added to it (or removed from the model SST) for an accurate comparison. This limits the interpretation of the PGM4 warm anomaly as a bias.
    Text modified at L310-314.
  - Turbulent heat flux differences could be substantial between the "flux" and "bulk formulations" and explain mean SST differences between PGM4 and GULF18.
    Text modified at L479-480
  - Solar radiation and background TKE differences are inconsistent with a warmer SST in PGM4 compared to GULF18: they would tend to warm the water column at higher depths.
    Text modified at L486-487

- I feel that Table 3,4,5,6 and Fig.11 could be made simpler and statistically more robust by merging all years in one single dataset.
  - Tab3 changed as suggested. Text consistently amended at L460-461.
  - Tab4 and Fig11 changed as suggested. Text consistently amended at L500-502, L509, L517-520, L528.
  - Tab5 is left as it is since the areas are different between the 2014 and 2017 and averaging is not possible.

- o Tab6 is changed as suggested. Text consistently amended at L597-600, L619-620.